# GFlowNet Assisted Biological Sequence Editing

**Pouya M. Ghari**[*]
University of California Irvine

**Alex M. Tseng**
Genentech

**Gökcen Eraslan**
Genentech

**Romain Lopez**
Genentech, Stanford University

**Tommaso Biancalani**
Genentech

**Gabriele Scalia**
Genentech

**Ehsan Hajiramezanali**[†]
Genentech

## Abstract

Editing biological sequences has extensive applications in synthetic biology and medicine, such as designing regulatory elements for nucleic-acid therapeutics and treating genetic disorders. The primary objective in biological-sequence editing is to determine the optimal modifications to a sequence which augment certain biological properties while adhering to a minimal number of alterations to ensure predictability and potentially support safety. In this paper, we propose GFNSeqEditor, a novel biological-sequence editing algorithm which builds on the recently proposed area of generative flow networks (GFlowNets). Our proposed GFNSeqEditor identifies elements within a starting seed sequence that may compromise a desired biological property. Then, using a learned stochastic policy, the algorithm makes edits at these identified locations, offering diverse modifications for each sequence to enhance the desired property. The number of edits can be regulated through specific hyperparameters. We conducted extensive experiments on a range of real-world datasets and biological applications, and our results underscore the superior performance of our proposed algorithm compared to existing state-of-the-art sequence editing methods.

## 1 Introduction

Editing biological sequences has a multitude of applications in biology, medicine, and biotechnology. For instance, gene editing serves as a tool to elucidate the role of individual gene products in diseases [28] and offers the potential to rectify genetic mutations in afflicted tissues and cells for therapeutic interventions [9]. The primary objective in biological-sequence editing is to enhance specific biological attributes of a starting seed sequence, while minimizing the number of edits. This reduction in the number of alterations not only has the potential to improve safety but also facilitates the predictability and precision of modification outcomes.

Existing methodologies that leverage generative modeling in the context of biological sequences have predominantly concentrated on *de novo* generation of sequences with desired properties [52, 61, 3]. A common feature of these approaches is generating entirely new sequences from scratch. As a result, there is an inherent risk of deviating significantly from naturally occurring sequences, compromising safety (e.g., the risk of designing sequences that might trigger an immune response) and predictability (e.g., obtaining misleading predictions from models that are trained on genomic sequences due to

---

[*]Work has been done while interning at Genentech
[†]Corresponding author: `hajiramezanali.ehsan@gene.com`.

38th Conference on Neural Information Processing Systems (NeurIPS 2024).

out-of-distribution effects). Despite the paramount importance of editing biological sequences, there has been a noticeable scarcity of research leveraging generative modeling to address this aspect specifically.

Generative flow networks (GFlowNets) [5, 6], a generative approach recognized for their ability to sequentially generate new objects, have shown remarkable performance in generating novel biological sequences from scratch [19, 30]. Drawing inspiration from the emerging field of GFlowNets, this paper introduces a novel biological-sequence editing algorithm: *GFNSeqEditor*. GFNSeqEditor assesses the potential for significant property enhancement within a given seed sequence by iteratively identifying and subsequently editing specific positions in the input sequence. More precisely, using the trained flow function, GFNSeqEditor first identifies positions in the seed sequence that require editing. Then, it constructs a stochastic policy using the flow function to select a substitution from the available options for the identified positions. Our stochastic approach empowers GFNSeqEditor to generate a diverse set of edited sequences for each input sequence, which, due to the diverse nature of biological targets, is an important consideration in biological sequence design [34, 19].

In summary, this paper makes the following contributions:

- We introduce GFNSeqEditor, a novel sequence-editing method which identifies and edits positions within a given sequence. GFNSeqEditor generates diverse edits for each input sequence based on a stochastic policy.

- We theoretically analyze the properties of the sequences edited through GFNSeqEditor, deriving lower and upper bounds on the property of edited sequences. Additionally, we demonstrate that the lower and upper bounds for the number of edits performed by GFNSeqEditor can be controlled through the adjustment of hyperparameters (Subsection 4.3).

- We conduct experiments across various DNA and protein sequence editing tasks, showcasing GFNSeqEditor's remarkable efficiency in enhancing properties with a reduced number of edits when compared to existing state-of-the-art methods. (Subsection 5.1).

- We highlight the versatility of GFNSeqEditor, which can be employed not only for sequence editing but also alongside biological-sequence generation models to produce novel sequences with improved properties and increased diversity (Subsection 5.2).

- We demonstrate the usage of GFNSeqEditor for sequence length reduction, allowing the creation of new, relatively shorter sequences by combining pairs of long and short sequences (Subsection 5.3).

## 2  Related Works

**De Novo Sequence Design.** The generation of biological sequences has been tackled using a diverse range of methods, including reinforcement learning [1], Bayesian optimization [51], deep generative models for search and sampling [18], generative adversarial networks [61], diffusion models [3], model-based optimization approaches [52, 7], adaptive evolutionary strategies [16, 49], likelihood-free inference [55], and surrogate-based black-box optimization [10], and GFlowNet [19]. It is important to note that all these sequence-generation methods generate sequences from scratch. However, *ab initio* generation carries the risk of *deviating too significantly* from naturally occurring sequences, which can compromise safety and predictability. In contrast, our proposed method enhances a target property while maintaining the similarity to seed sequences (e.g., naturally occurring sequences), thus improving predictability and potentially enhancing safety.

**Sequence Editing.** Traditional approaches commonly employed for biological sequence editing are evolution-based methods, where—over many iterations—a starting "seed" sequence is randomly mutated, retaining only the best sequences (i.e., highest desired property) for the next round [2, 46, 50, 41]. These approaches have several important limitations. First, they require the evaluation of numerous candidate sequences at every iteration. This computational demand can become prohibitively expensive, particularly for lengthy sequences. Additionally, evolution-based methods heavily rely on evaluations provided by a proxy model capable of assessing the properties of unseen sequences; the efficacy of these methods is thus limited by the reliability of the underlying proxy. Moreover, these methods may require repeated rounds of interactions with the lab [41], which can be costly and time-consuming.

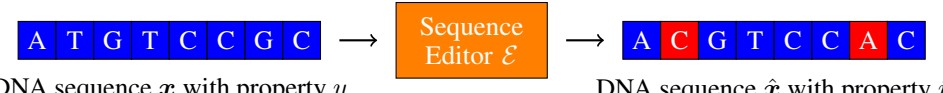

DNA sequence $\boldsymbol{x}$ with property $y$            DNA sequence $\hat{\boldsymbol{x}}$ with property $\hat{y}$

Figure 1: An example of editing the DNA sequence 'ATGTCCGC'. The goal is to make a limited number of edits to maximize the property $\hat{y}$. Each token in the sequence in this example is called a *base* and can be any of ['A', 'C', 'T', 'G']. The editor function $\mathcal{E}$ accepts the initial sequence as an input and determines that the second and seventh bases require editing (highlighted in red). Then, $\mathcal{E}$ modifies the bases at these identified locations to improve the property value.

Beyond evolution-based methods, a handful of optimization-based methods have been proposed by [42, 48, 21]. By treating sequence editing as an optimization task, Ledidi [42] learns to perturb specific positions within a given sequence. Utilizing Bayesian optimization, LaMBO [48] generates new sequences by optimizing a batch of starting seed sequences. Building upon the LaMBO framework, MOGFN-AL [21] leverages GFlowNets to generate candidates in each round of Bayesian optimization loop, improving computational efficiency compared to LaMBO. Akin to evolution-based models, these optimization-based methods require the evaluation of unseen sequences. Consequently, their effectiveness is contingent on the quality of the proxy model, which can compromise their performance if the proxy model lacks sufficient generalizability for unseen sequences. Furthermore, both evolution-based and optimization-based methods perform local searches given either a single seed sequence or a batch of seed sequences. Thus, these methods face issues related to low sample efficiency. In contrast, GFNSeqEditor relies on a pre-trained flow function that amortizes the search cost over the learning process, allocating probability mass across the entire space to facilitate exploration and diversity. Furthermore, GFNSeqEditor can be employed for editing without necessitating the evaluation of unseen sequence properties. Theoretical analysis presented in this paper establishes that the bounds of edited sequence rewards, property improvement, and the number of edits can be effectively regulated through GFNSeqEditor hyperparameters. Therefore, GFNSeqEditor offers increased reliability and operational suitability in comparison to counterparts lacking a robust theoretical analysis.

We provide an extensive overview of the related literature, with additional discussion available in the Appendix F.

## 3 Preliminaries and Problem Statement

Let $\boldsymbol{x}$ be a biological sequence with property $y$. For example, $\boldsymbol{x}$ may be a DNA sequence, and $y$ may be the likelihood it binds to a particular protein of interest. The present paper considers the problem of searching for edits in $\boldsymbol{x}$ to improve $y$. To this end, the goal is to learn an editor function $\mathcal{E}(\cdot)$ which accepts a sequence $\boldsymbol{x}$ and outputs the edited sequence $\mathcal{E}(\boldsymbol{x}) = \hat{\boldsymbol{x}}$ with property $\hat{y}$. The editor function $\mathcal{E}(\cdot)$ should maximize $\hat{y}$, while at the same time minimizing the number of edits between $\boldsymbol{x}$ and $\hat{\boldsymbol{x}}$. To achieve this goal, we propose GFNSeqEditor. GFNSeqEditor first identifies positions in a given biological sequence such that editing those positions leads to considerable improvement in the property of the sequence. Then, the learned editor function $\mathcal{E}$ edits these identified locations (Figure 1). GFNSeqEditor uses a trained GFlowNet [5, 6] to identify positions that require editing and subsequently generate edits for those positions. The following Subsections present preliminaries on GFlowNets.

### 3.1 Generative Flow Networks

Generative Flow Networks (GFlowNets) [5, 6] learn a stochastic policy $\pi(\cdot)$ to sequentially construct a discrete object $\boldsymbol{x}$. Let $\mathcal{X}$ be the space of discrete objects $\boldsymbol{x}$. It is assumed that the space $\mathcal{X}$ is compositional, meaning that an object $\boldsymbol{x}$ can be constructed using a sequence of actions taken from an action set $\mathbb{A}$. At each step $t$, given a partially constructed object $\boldsymbol{s}_t$, GFlowNet samples an action $a_{t+1}$ from the set $\mathbb{A}$ using the stochastic policy $\pi(\cdot|\boldsymbol{s}_t)$. Then, GFlowNet appends $a_{t+1}$ to $\boldsymbol{s}_t$ to obtain $\boldsymbol{s}_{t+1}$. In this context, $\boldsymbol{s}_t$ can be viewed as the state at step $t$. The above procedure continues until reaching a terminating state, which yields the fully constructed object $\boldsymbol{x}$. To construct an object $\boldsymbol{x}$, the GFlowNet starts from an initial empty state $\boldsymbol{s}_0$, and applying actions sequentially, all fully constructed objects must end in a special final state $\boldsymbol{s}_f$. Therefore, the trajectory of states to construct

an object $\boldsymbol{x}$ can be written as $\tau_{\boldsymbol{x}} = (\boldsymbol{s}_0 \rightarrow \boldsymbol{s}_1 \rightarrow \cdots \rightarrow \boldsymbol{x} \rightarrow \boldsymbol{s}_f)$. Let $\mathbb{T}$ be the set of all possible trajectories. Furthermore, let $R(\cdot) : \mathcal{X} \rightarrow \mathbb{R}^+$ be a non-negative reward function defined on $\mathcal{X}$. The goal of GFlowNet is to learn a stochastic policy $\pi(\cdot)$ such that $\pi(\boldsymbol{x}) \propto R(\boldsymbol{x})$. This means that the GFlowNet learns a stochastic policy $\pi(\cdot)$ to generate an object $\boldsymbol{x}$ with a probability proportional to its reward.

As described later, to obtain the policy $\pi(\cdot)$, the GFlowNet uses trajectory flow $F : \mathbb{T} \rightarrow \mathbb{R}^+$. The trajectory flow $F(\tau)$ assigns a probability mass to the trajectory $\tau$. Then, the *edge flow* from state $\boldsymbol{s}$ to state $\boldsymbol{s}'$ is defined as $F(\boldsymbol{s} \rightarrow \boldsymbol{s}') = \sum_{\forall \tau : \boldsymbol{s} \rightarrow \boldsymbol{s}' \in \tau} F(\tau)$. Moreover, the *state flow* is defined as $F(\boldsymbol{s}) = \sum_{\forall \tau : \boldsymbol{s} \in \tau} F(\tau)$. The trajectory flow $F(\cdot)$ induces a probability measure $P_F(\cdot)$ over completed trajectories that can be expressed as $P_F(\tau) = \frac{F(\tau)}{Z}$ where $Z = \sum_{\forall \tau \in \mathbb{T}} F(\tau)$ represents the total flow. The probability of visiting state $\boldsymbol{s}$ can be written as $P_F(\boldsymbol{s}) = \frac{\sum_{\forall \tau \in \mathbb{T} : \boldsymbol{s} \in \tau} F(\tau)}{Z}$. Then, the forward transition probability from state $\boldsymbol{s}$ to state $\boldsymbol{s}'$ can be obtained as $P_F(\boldsymbol{s}'|\boldsymbol{s}) = \frac{F(\boldsymbol{s} \rightarrow \boldsymbol{s}')}{F(\boldsymbol{s})}$. The trajectory flow $F(\cdot)$ is called a consistent flow if for any state $\boldsymbol{s}$ it satisfies $\sum_{\forall \boldsymbol{s}' : \boldsymbol{s}' \rightarrow \boldsymbol{s}} F(\boldsymbol{s}' \rightarrow \boldsymbol{s}) = \sum_{\forall \boldsymbol{s}'' : \boldsymbol{s} \rightarrow \boldsymbol{s}''} F(\boldsymbol{s} \rightarrow \boldsymbol{s}'')$, which constitutes that the in-flow and out-flow of state $\boldsymbol{s}$ are equal. [5] shows that if $F(\cdot)$ is a consistent flow such that the terminal flow is set as reward (i.e. $F(\boldsymbol{x} \rightarrow \boldsymbol{s}_f) = R(\boldsymbol{x})$), the policy $\pi(\cdot)$ defined as $\pi(\boldsymbol{s}'|\boldsymbol{s}) = P_F(\boldsymbol{s}'|\boldsymbol{s})$ satisfies $\pi(\boldsymbol{x}) = \frac{R(\boldsymbol{x})}{Z}$ which means that the policy $\pi(\cdot)$ samples an object $\boldsymbol{x}$ proportional to its reward.

## 3.2 Training GFlowNet Models

In order to learn the policy $\pi(\cdot)$, a GFlowNet model approximates trajectory flow with a flow function $F_{\boldsymbol{\theta}}(\cdot)$ where $\boldsymbol{\theta}$ includes learnable parameters of the flow function. To learn the flow function that can provide consistency condition, [5] formulates flow-matching loss function as follows:

$$\mathcal{L}_{\text{FM}}(\boldsymbol{s}; \boldsymbol{\theta}) = \left( \log \frac{\sum_{\forall \boldsymbol{s}' : \boldsymbol{s}' \rightarrow \boldsymbol{s}} F_{\boldsymbol{\theta}}(\boldsymbol{s}' \rightarrow \boldsymbol{s})}{\sum_{\forall \boldsymbol{s}'' : \boldsymbol{s} \rightarrow \boldsymbol{s}''} F_{\boldsymbol{\theta}}(\boldsymbol{s} \rightarrow \boldsymbol{s}'')} \right)^2 . \tag{1}$$

Moreover, as an alternative objective function, [31] introduces trajectory balance as:

$$\mathcal{L}_{\text{TB}}(\boldsymbol{s}; \boldsymbol{\theta}) = \left( \log \frac{Z_{\boldsymbol{\theta}} \prod_{\boldsymbol{s} \rightarrow \boldsymbol{s}'} P_{F_{\boldsymbol{\theta}}}(\boldsymbol{s}'|\boldsymbol{s})}{R(\boldsymbol{x})} \right)^2 \tag{2}$$

where $Z_{\boldsymbol{\theta}}$ is a learnable parameter. The trajectory-balance objective function in (2) can accelerate training GFlowNets and provide robustness to long trajectories. Given a training dataset, optimization techniques such as stochastic gradient descent can be applied to objective functions in (1) and (2) to train the GFlowNet model. We use trajectory balance in this paper due to its well-documented performance. Furthermore, it is worth noting that generating sequences in an autoregressive fashion using GFlowNet involves only one path to generate a particular sequence. In such cases, generating biological sequences with GFlowNet can be viewed as a Soft-Q-Learning [15, 13, 33] and path consistency learning (PCL) [35] problem.

## 4 Sequence Editing with GFlowNet

To edit a given sequence $\boldsymbol{x}$, we propose identifying *sub-optimal* positions of $\boldsymbol{x}$ such that editing them can lead to considerable improvement in the sequence property. Assume that the flow function $F_{\boldsymbol{\theta}}(\cdot)$ is trained on available offline training data. GFNSeqEditor uses the trained GFlowNet's flow function $F_{\boldsymbol{\theta}}(\cdot)$ to identify sub-optimal positions of $\boldsymbol{x}$, and subsequently replace the sub-optimal parts with newly sampled edits based on the stochastic policy $\pi(\cdot)$.

### 4.1 Sub-Optimal-Position Identification

This Subsection provides intuition on how GFNSeqEditor uses a pre-trained flow function $F_{\boldsymbol{\theta}}(\cdot)$ to identify sub-optimal positions in a sequence $\boldsymbol{x}$ to edit. Let $x_t$ and $\boldsymbol{x}_{:t}$ denote the $t$-th element and the first $t$ elements in the sequence $\boldsymbol{x}$, respectively. For example, in the DNA sequence $\boldsymbol{x} = $ 'ATGTCCGC', we have $x_2 = $ 'T' and $\boldsymbol{x}_{:2} = $ 'AT'. GFNSeqEditor constructs edited sequences token by token, and for each position $t + 1$, it examines whether $x_{t+1}$ should be edited or not. Using the flow function $F_{\boldsymbol{\theta}}(\cdot)$, given $\boldsymbol{x}_{:t}$, GFlowNet evaluates the average reward obtained by appending any possible token

to $\boldsymbol{x}_{:t}$. In this context, each token can be viewed as an action. Let $\boldsymbol{x}_{:t} + a$ denotes the expanded $\boldsymbol{x}_{:t}$ by appending token $a$. Let $\mathbb{A}$ represent the available action set. For each $a \in \mathbb{A}$, using the state flow $F_{\boldsymbol{\theta}}(\boldsymbol{x}_{:t} + a)$, the value of action $a$ given $\boldsymbol{x}_{:t}$ can be evaluated. As discussed in Section 3, the state flow $F_{\boldsymbol{\theta}}(\boldsymbol{x}_{:t} + a)$ is proportional to the total reward of all possible sequences that have $\boldsymbol{x}_{:t} + a$ as their prefix. Therefore, $F_{\boldsymbol{\theta}}(\boldsymbol{x}_{:t} + a_1) > F_{\boldsymbol{\theta}}(\boldsymbol{x}_{:t} + a_2)$ indicates that taking action $a_1$ instead of action $a_2$ can lead to obtaining better candidates for the final sequence. We can leverage this property of the flow function $F_{\boldsymbol{\theta}}(\cdot)$ to examine if $x_{t+1}$ is sub-optimal or not. If the reward resulting from having $x_{t+1}$ in the seed sequence is evaluated by $F_{\boldsymbol{\theta}}(\cdot)$ to be relatively small compared to other possible actions, then $x_{t+1}$ is considered sub-optimal. In particular, $x_{t+1}$ is identified as sub-optimal if:

$$F_{\boldsymbol{\theta}}(\boldsymbol{x}_{:t} + x_{t+1}) < \delta \max_{a \in \mathbb{A}} F_{\boldsymbol{\theta}}(\boldsymbol{x}_{:t} + a), \tag{3}$$

where $0 \leq \delta \leq 1$ is a hyperparameter. A larger $\delta$ increases the likelihood that the algorithm identifies $x_{t+1}$ as sub-optimal. From (3), it can be inferred that $x_{t+1}$ is identified as sub-optimal if its associated out-flow is considerably smaller than the out-flow associated with the best possible action in $\mathbb{A}$. This means that the flow function $F_{\boldsymbol{\theta}}(\cdot)$ suggests that replacing $x_{t+1}$ with other actions can lead to remarkable improvement in the sequence property.

## 4.2 Sequence Editing with GFNSeqEditor

Using the flow function $F_{\boldsymbol{\theta}}(\cdot)$, GFNSeqEditor iteratively identifies and edits positions in a seed sequence. Subsection 4.1 presented a simple function for determining if a position $x_{t+1}$ in a sequence should be edited to improve the target property value ((3)). Based on this intuition, we now modify (3) to formally define the sub-optimal-position identification function $D(\cdot)$ used by GFNSeqEditor.

Let $\hat{\boldsymbol{x}}_{:t}$ denote the first $t$ elements of the edited sequence. Assume that $x_t \in \mathbb{A}$, $\forall t$, meaning that $x_t$ is always in the available actions. At each step $t$ of the algorithm, $D(\cdot)$ accepts $\hat{\boldsymbol{x}}_{:t-1}$ and evaluates whether appending $x_t$ (from the seed sequence) to the edited partial sequence $\hat{\boldsymbol{x}}_{:t-1}$ is detrimental to the performance. In particular, modifying (3), the sub-optimal identifier function $D(\cdot)$ checks the following condition:

$$\frac{F_{\boldsymbol{\theta}}(\hat{\boldsymbol{x}}_{:t-1} + x_t)}{\sum_{a' \in \mathbb{A}} F_{\boldsymbol{\theta}}(\hat{\boldsymbol{x}}_{:t-1} + a')} < \delta \max_{a \in \mathbb{A}} \frac{F_{\boldsymbol{\theta}}(\hat{\boldsymbol{x}}_{:t-1} + a)}{\sum_{a' \in \mathbb{A}} F_{\boldsymbol{\theta}}(\hat{\boldsymbol{x}}_{:t-1} + a')} + \nu, \tag{4}$$

where $\nu \sim \mathcal{N}(0, \sigma^2)$ is a Gaussian random variable with variance of $\sigma^2$. The variance $\sigma^2$ is a hyperparameter. The relation between $\sigma$ and the algorithm performance will be analyzed in Section 4.3. The inclusion of additive noise $\nu$ on the right-hand side of (4) introduces a degree of randomness into the process of identifying sub-optimal positions. This, in turn, fosters exploration in the editing process. The sub-optimal-position-identifier function $D(\cdot)$ determines if $x_t$ is sub-optimal as follows:

$$D(x_t, \hat{\boldsymbol{x}}_{:t-1}; \delta, \sigma) = \begin{cases} 1 & \text{If (4) is met} \\ 0 & \text{Otherwise} \end{cases}. \tag{5}$$

If $D(x_t, \hat{\boldsymbol{x}}_{:t-1}; \delta, \sigma) = 0$, at step $t$ the algorithm appends $x_t$ from the original sequence $\boldsymbol{x}$ to $\hat{\boldsymbol{x}}_{:t-1}$. Otherwise, if $D(x_t, \hat{\boldsymbol{x}}_{:t-1}; \delta, \sigma) = 1$, the algorithm samples an action $a$ according to the following policy:

$$\pi(a|\hat{\boldsymbol{x}}_{:t-1}) = (1 - \lambda) \frac{F_{\boldsymbol{\theta}}(\hat{\boldsymbol{x}}_{:t-1} + a)}{\sum_{a' \in \mathbb{A}} F_{\boldsymbol{\theta}}(\hat{\boldsymbol{x}}_{:t-1} + a')} + \lambda \mathbf{1}_{a=x_t}, \tag{6}$$

where $0 \leq \lambda < 1$ is a regularization coefficient and $\mathbf{1}_{a=x_t}$ denotes the indicator function and is 1 if $a = x_t$. The regularization parameter $\lambda$ allows tuning the sampling process to favor the original sequence (a larger $\lambda$ leads to a smaller number of edits). The policy in (6) constitutes a trade-off between increasing the target property and decreasing the distance between the edited sequence $\hat{\boldsymbol{x}}$ and the original sequence $\boldsymbol{x}$. Let $\tilde{x}_t$ be the action sampled by the policy $\pi$ in (6). In summary, the $t$-th element in the edited sequence can be written as:

$$\hat{x}_t = D(x_t, \hat{\boldsymbol{x}}_{:t-1}; \delta, \sigma) \tilde{x}_t + (1 - D(x_t, \hat{\boldsymbol{x}}_{:t-1}; \delta, \sigma)) x_t. \tag{7}$$

Therefore, at each step $t$, the edited sequence is updated as $\hat{\boldsymbol{x}}_{:t} = \hat{\boldsymbol{x}}_{:t-1} + \hat{x}_t$. The process continues until step $T$, where $T = |\boldsymbol{x}|$ denotes the length of the original sequence $\boldsymbol{x}$. Note that $\hat{\boldsymbol{x}}_{:0}$ is an empty sequence. Algorithm 1 summarizes GFNSeqEditor.

---

**Algorithm 1** GFNSeqEditor: Sequence Editor using GFlowNet

---
1: **Input:** Sequence $\boldsymbol{x}$ with length $T$, flow function $F_{\boldsymbol{\theta}}(\cdot)$ and parameters $\delta$, $\lambda$ and $\sigma$.
2: Initialize $\hat{\boldsymbol{x}}_{:0}$ as an empty sequence.
3: **for** $t = 1, \ldots, T$ **do**
4:     Check if $x_t$ is sub-optimal by obtaining $D(x_t, \hat{\boldsymbol{x}}_{:t-1}; \delta, \sigma)$ according to (5).
5:     **if** $D(\hat{\boldsymbol{x}}_{:t-1}; \delta, \sigma) = 1$ **then**
6:         Sample $\hat{x}_t$ according to policy $\pi(\cdot|\hat{\boldsymbol{x}}_{:t-1})$ in (6).
7:     **else**
8:         Assign $\hat{x}_t = x_t$.
9:     **end if**
10: **end for**
11: **Output:** Edited sequence $\hat{\boldsymbol{x}}$.

---

## 4.3 Analysis

This Subsection analyzes the reward and properties of the edited sequence as well as the number of edits performed by GFNSeqEditor. Specifically, the bounds for the reward of edited sequences, property improvement and the number of edits are determined by the algorithm's hyperparameters $\sigma$, $\delta$, and $\lambda$. The following theorem specifies the lower bound for the reward of edited sequences.

**Theorem 4.1.** *Let $T$ be the length of the original sequence $\boldsymbol{x}$. The expected reward of the sequence edited by GFNSeqEditor $\hat{\boldsymbol{x}}$ given $\boldsymbol{x}$ is bounded from below as:*

$$\mathbb{E}[R(\hat{\boldsymbol{x}})|\boldsymbol{x}] \geq \left(1 - \Phi(\frac{1-\delta}{\sigma})\right)(1-\lambda)R_{F,T}, \tag{8}$$

*where $\Phi(\cdot)$ denotes the cumulative distribution function (CDF) for the normal distribution and $R_{F,T}$ represents the expected reward of a sequence with length $T$ generated using the flow function $F_{\boldsymbol{\theta}}(\cdot)$.*

Proof of Theorem 4.1 is deferred to Appendix A. The following Theorem obtains the expected property improvement upper bound of the proposed GFNSeqEditor. The property improvement of a sequence $\boldsymbol{x}$ is defined as $\text{PI} = \hat{y} - y$ where $\hat{y}$ denotes the edited sequence property.

**Theorem 4.2.** *Let $\mathbb{S}_{\boldsymbol{x}}^*$ be the set of all sequences with length $T$ which have larger properties than that of $\boldsymbol{x}$ (i.e., $y$). Assume that $\mathbb{S}_{\boldsymbol{x}}^*$ is a non-empty set. The expected property improvement by applying GFNSeqEditor on $\boldsymbol{x}$ is bounded from above as*

$$\mathbb{E}[PI|\boldsymbol{x}] \leq \sum_{\boldsymbol{w} \in \mathbb{S}_{\boldsymbol{x}}^*} \left(1 - \Phi(-\frac{\delta}{\sigma})\right)(p_{\boldsymbol{w}} - y) \tag{9}$$

*where $p_{\boldsymbol{w}}$ denote the property of the sequence $\boldsymbol{w}$.*

The proof of Theorem 4.2 can be found in Appendix B. Theorems 4.1 and 4.2 demonstrate that an increase in $\delta$ results in an increase in both the lower bound of reward and the upper bound of property improvement. While a higher value of $\sigma$ corresponds to larger lower bounds for the reward, an increase in $\sigma$ diminishes the upper bound of the property improvement. The following theorem obtains the upper bound on the number of edits performed by the proposed GFNSeqEditor.

**Theorem 4.3.** *The expected distance between the edited sequence $\hat{\boldsymbol{x}}$ by GFNSeqEditor and the original sequence $\boldsymbol{x}$ is bounded from above as:*

$$\mathbb{E}[\text{lev}(\boldsymbol{x}, \hat{\boldsymbol{x}})] \leq \left[(1-\lambda)\left(1 - \Phi(-\frac{\delta}{\sigma})\right)\right]T, \tag{10}$$

*where $\text{lev}(\cdot, \cdot)$ is the Levenshtein distance between two sequences.*

The proof for Theorem 4.3 is available in Appendix C. The following Theorem specifies the lower bound for the number of edits.

**Theorem 4.4.** *Let there exists $\epsilon > 0$ such that the flow function $F_{\boldsymbol{\theta}}(\cdot)$ satisfies:*

$$\max_{a \in \mathbb{A}} \frac{F_{\boldsymbol{\theta}}(\hat{\boldsymbol{x}}_{:t-1} + a)}{\sum_{a' \in \mathbb{A}} F_{\boldsymbol{\theta}}(\hat{\boldsymbol{x}}_{:t-1} + a')} \leq 1 - \epsilon, \forall t, \tag{11}$$

*meaning that the probability of choosing of each action is always less than $1 - \epsilon$. The expected distance between the edited sequence $\hat{x}$ by GFNSeqEditor and the original sequence $x$ is bounded from below as:*

$$\mathbb{E}[\text{lev}(\boldsymbol{x}, \hat{\boldsymbol{x}})] \geq \left[ \epsilon(1 - \lambda) \left( 1 - \Phi(\frac{1 - \delta}{\sigma}) \right) \right] T. \tag{12}$$

Proof of Theorem 4.4 can be found in Appendix D. Theorems 4.3 and 4.4 show that as $\delta$ increases, both the lower and upper bounds of distance increase. In contrast, an increase in $\lambda$ leads to a decrease in both the lower and upper bounds of distance. Furthermore, Theorem 4.1 demonstrates that a reduction in $\lambda$ results in a larger lower bound for the reward. Therefore, Theorems 4.1 and 4.3 reveal a trade-off between the expected number of edits and the lower bound for the expected reward. While it is preferable to select hyperparameters $\delta$ and $\lambda$ that reduce the expected number of edits, an increase in the number of edits corresponds to a larger lower bound for the reward.

## 5 Experiments

We conducted extensive experiments to assess the performance of GFNSeqEditor in comparison to several state-of-the-art baselines across diverse DNA- and protein-sequence editing tasks. We evaluate on TFbinding, AMP, and CRE datasets. TFbinding and CRE datasets consist DNA sequences with lengths of 8 and 200, respectively. The task in both datasets is to edit sequences to increase their binding activities. The vocabulary for both TFbinding and CRE is the four DNA bases, {A, C, G, T}. AMP dataset comprises positive samples, representing anti-microbial peptides (AMPs), and negative samples, which are non-AMPs. The vocabulary consists of 20 amino acids. The primary objective is to edit the non-AMP samples in such a way that the edited versions attain the characteristics exhibited by AMP samples. Additional information about the datasets can be found in Appendix E.1.1.

To evaluate the performance of sequence editing methods, we compute the following metrics:

- **Property Improvement (PI):** The PI for a given sequence $\boldsymbol{x}$ with label $y$ is calculated as the average enhancement in property across edits, expressed as PI $= \frac{1}{n_e} \sum_{i=1}^{n_e} (\hat{y}_i - y)$, where $n_e$ is the number of edited sequences associated with the original sequence $\boldsymbol{x}$ and $\hat{y}_i$ denote the property of the $i$-th edited sequence $\hat{\boldsymbol{x}}_i$. To evaluate the performance of editing methods, for each dataset we leverage an oracle to obtain $\hat{y}_i$ given $\hat{\boldsymbol{x}}_i$. More details about oracles can be found in Appendix E.

- **Edit Percentage (EP):** The average Levenshtein distance between $\boldsymbol{x}$ and edited sequences normalized by the length of $\boldsymbol{x}$ expressed as $\frac{1}{n_e T} \sum_{i=1}^{n_e} \text{lev}(\boldsymbol{x}, \hat{\boldsymbol{x}}_i)$.

- **Diversity:** For each sequence $\boldsymbol{x}$, the diversity among edited sequences can be obtained as $\frac{2}{n_e(n_e-1)} \sum_{i=1}^{n_e-1} \sum_{j=i+1}^{n_e} \text{lev}(\hat{\boldsymbol{x}}_i, \hat{\boldsymbol{x}}_j)$.

- **GMDPI:** The geometric mean of diversity and PI is measured. This metric highlights algorithms that exhibit strong performance in both aspects simultaneously.

We compared GFNSeqEditor to several baselines, including Directed Evolution (DE) [46], Ledidi [42], LaMBO [48], MOGFN-AL [21], GFlowNet-AL [19], and Seq2Seq. To perform Directed Evolution for sequence editing, we select a set of positions uniformly at random within a given sequence and then apply the directed-evolution algorithm to edit these positions. The implementation of the directed-evolution algorithm is the same as that of the AdaLead framework in [46]. Inspired by graph-to-graph translation for molecular optimization in [23], we implemented another editing baseline, which is called Seq2Seq. For the Seq2Seq baseline, we initially partition the dataset into two subsets: i) sequences with lower target-property values, and ii) sequences with relatively higher target-property values. Subsequently, we create pairs of data samples such that each low-property sequence is paired with its closest counterpart from the high-property sequence set, based on Levenshtein distance. A transformer is then trained to map each low-property sequence to its high-property pair. Essentially, the Seq2Seq baseline maps an input sequence to a similar sequence with a higher property value. Furthermore, we adapted GFlowNet-AL for sequence editing, and named it GFlowNet-E in what follows. In this baseline, the initial segment of the sequence serves as the input, allowing the model to generate the subsequent portion of the sequence. For TF-binding, AMP, and CRE datasets, GFlowNet-E takes in the initial 70%, 65%, and 60% of elements, respectively, from the

Table 1: Performance of GFNSeqEditor compared to the baselines in terms of property improvement (PI), edit percentage (EP), diversity, and geometric mean of property improvement and diversity (GMDPI) on TFbinding, AMP, and CRE datasets. EP is selected to be approximately the same for all algorithms (if possible). Higher PI, diversity and GMDPI are preferable.

| Methods | TFbinding | | | | AMP | | | | CRE | | | |
|---|---|---|---|---|---|---|---|---|---|---|---|---|
| | PI | EP(%) | Diversity | GMDPI | PI | EP(%) | Diversity | GMDPI | PI | EP(%) | Diversity | GMDPI |
| DE | 0.12 | 25.00 | 3.01 | 0.60 | 0.11 | 33.82 | 13.67 | 1.23 | 0.63 | 22.93 | 62.07 | 6.25 |
| Ledidi | 0.06 | 27.80 | 1.25 | 0.27 | 0.18 | 34.79 | 11.65 | 1.45 | 1.36 | 22.13 | 50.49 | 8.29 |
| LaMBO | 0.05 | 25.00 | 3.14 | 0.40 | 0.12 | 34.33 | **15.61** | 1.36 | 0.79 | 23.35 | **62.95** | 7.05 |
| MOGFN-AL | 0.09 | 25.00 | 2.66 | 0.49 | 0.10 | 35.26 | 7.59 | 0.87 | 2.45 | 22.99 | 10.96 | 5.18 |
| GFlowNet-E | 0.11 | 28.35 | 2.10 | 0.48 | 0.28 | 35.68 | 3.42 | 0.98 | 4.24 | 22.73 | 37.06 | 12.53 |
| Seq2Seq | 0.03 | 41.98 | - | - | 0.21 | 78.05 | - | - | - | - | - | - |
| GFNSeqEditor | **0.14** | 24.27 | **3.84** | **0.73** | **0.33** | 34.49 | 14.34 | **2.17** | **9.90** | 21.90 | 40.41 | **20.00** |

input sequence $x$, and generates the remaining elements using the pre-trained flow function. More details on the baselines can be found in Appendix E.1.

To train both the baselines and the proposed GFNSeqEditor, we divide each dataset into training, validation, and test sets with proportions of $72\%$, $18\%$ and $10\%$, respectively. The test set serves the purpose of evaluating the performance of the methods in sequence editing tasks. The flow function $F_\theta(\cdot)$ utilized by GFNSeqEditor and the GFlowNet-E baseline is an MLP consisting of two hidden layers, each with a dimension of 2048, and $|\mathbb{A}|$ outputs corresponding to actions. Throughout our experiments, we employ the trajectory balance objective to train the flow function. Additional details regarding the training of the flow function can be found in Appendix E.1.

## 5.1 Sequence Editing

Table 1 presents the performance of GFNSeqEditor and other baselines on TFbinding, AMP, and CRE datasets[3]. We set GFNSeqEditor and all the baselines except for Seq2Seq to create 10 edited sequences for each input sequence. The Seq2Seq implementation closely resembles a deterministic machine translator and is limited to producing just one edited sequence per input, resulting in a diversity

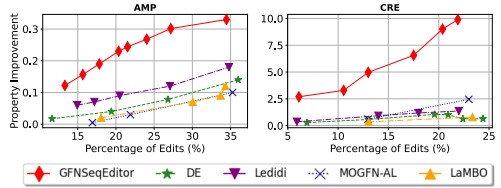

Figure 2: Property improvement of AMP **(left)** and CRE **(right)** with respect to edit percentage.

score of zero. Additionally, Figure 2 shows the property improvement achieved by GFNSeqEditor, DE, Ledidi, LaMBO, and MOGFN-AL across a range of edit percentages. As evident from Table 1 and Figure 2, GFNSeqEditor outperforms all baselines, achieving substantial property improvements with a controlled number of edits. This superior performance is attributed to GFNSeqEditor's utilization of a pre-trained flow function from GFlowNet, enabling it to achieve significantly higher property improvements than DE, Ledidi, LaMBO, and MOGFN-AL, which rely on local search techniques by optimizing either a given single sequence or a batch of sequences. Specifically, the flow function $F_\theta(\cdot)$ is trained to sample sequences with probability proportional to their reward and, as a result, employing the policy in (6) for editing enables GFNSeqEditor to leverage global information contained in $F_\theta(\cdot)$ about the entire space of sequences. Furthermore, GFNSeqEditor achieves larger property improvement than GFlowNet-E. The GFNSeqEditor identifies and edits sub-optimal positions within a seed sequence using (4), while GFlowNet-E only edits the tail of the input seed sequence. This indicates the effectiveness of the sub-optimal position identifier function of GFNSeqEditor.

**Ablation study.** We further study the property improvement achieved by GFNSeqEditor along with edit percentage across various choices of hyperparameters $\delta$ and $\lambda$. Figure 3 illustrates that an increase in $\delta$ generally corresponds to an increase in both property improvement and edit percentage, whereas, in most cases, an increase in $\lambda$ results in a decrease in property improvement and edit percentage. Furthermore, in Figure 7 in Appendix E.3, we illustrate the impact of changing $\sigma$ on property improvement and edit diversity for GFNSeqEditor. This figure highlights that increasing $\sigma$ results in decreased property improvement and enhanced diversity. These results corroborate the theoretical analyses outlined in Theorems 4.1, 4.2 and 4.3 in Section 4.3.

---

[3]Seq2Seq relies on identifying pairs of similar sequences for training. However, we were unable to identify similar pairs for CRE, possibly because of the limited number of training samples relative to the lengthy nature of the sequences (i.e., sequences with a length of 200).

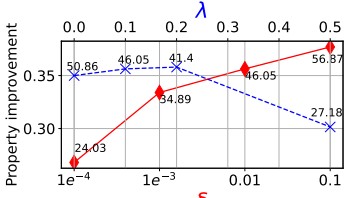 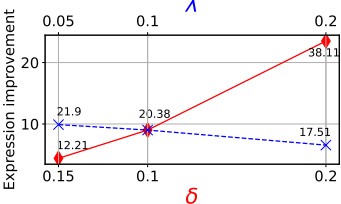

Figure 3: Studying the effect of heyperparameters $\delta$ and $\lambda$ on the performance of GFNSeqEditor over AMP **(left)** and CRE **(right)** datasets. The marker values are edit percentages.

## 5.2 Assisting Sequence Generation

In addition to editing sequences, we investigate the ability of GFNSeqEditor to be used alongside a sequence generative model to enhance the generation of novel sequences. This highlights the versatility of the proposed GFNSeqEditor. In this Subsection, we utilize a pre-trained diffusion model (DM) for sequence generation, with further details available in Appendix E.2. The sequences generated by the

Table 2: Performance of DM, GFlowNet and combination of DM with GFNSeqEditor for generating novel sequences.

| Algorithms | AMP | | CRE | |
|---|---|---|---|---|
| | Property | Diversity | Property | Diversity |
| DM | 0.66 | 23.86 | 1.75 | 107.38 |
| GFlowNet | 0.74 | 17.86 | 28.20 | 83.88 |
| DM+GFNSeqEditor | 0.73 | 23.78 | 26.42 | 103.10 |

DM are passed to GFNSeqEditor to improve their target property. Given that GFNSeqEditor utilizes a trained GFlowNet model, this combination of a DM and GFNSeqEditor can be regarded as an *ensemble approach*, effectively leveraging both the DM and the GFlowNet for sequence generation. Table 2 presents the property and diversity metrics for sequences generated by the DM, the GFlowNet, and the combined DM+GFNSeqEditor across AMP and CRE datasets, with each method generating 1,000 sequences. As observed from Table 2, GFlowNet excels at producing sequences with higher property values compared to the DM, while the DM exhibits greater sequence diversity than the GFlowNet. Sequences generated by DM+GFNSeqEditor maintain similar property levels to the GFlowNet on its own, while their diversity is in line with that of the DM. This highlights the effectiveness of DM+GFNSeqEditor in harnessing the benefits of both the GFlowNet and the DM.

Moreover, we show the CDF of the property for sequences generated by the DM, the GFlowNet, and DM+GFNSeqEditor in Figure 4. As shown, the CDF of DM+GFNSeqEditor aligns with both DM and GFlowNet. Specifically, for AMP dataset, DM+GFNSeqEditor generates more sequences with higher properties than 0.78 compared to GFlowNet, while reducing the number of low-property generated sequences compared to DM alone. In the case of CRE dataset, the

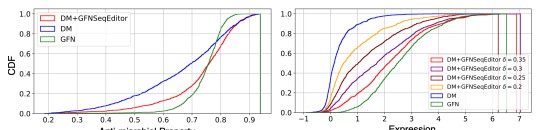

Figure 4: CDF of generated sequence properties for AMP **(left)** and CRE **(right)**. A right-shifted curve indicates that the model is generating more sequences that are high in the target property.

results in Figure 4 indicate that as $\delta$ increases, the CDF of DM+GFNSeqEditor becomes more akin to that of GFlowNet. This is expected, as an increase in $\delta$ leads to a greater number of edits.

## 5.3 Sequence Combination

GFNSeqEditor possesses the capability to combine multiple sequences, yielding a novel sequence that closely resembles its *parent* sequences. This capability proves invaluable in several applications. For example, when it is important to shorten relatively lengthy sequences while retaining desired properties (see, e.g., [54, 59]). GFNSeqEditor accomplishes this

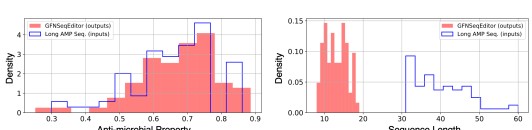

Figure 5: GFNSeqEditor effectively reduces the length of AMP sequence inputs **(right)** while keeping their properties intact **(left)**.

by combining a longer sequence with a shorter one. The resultant sequence maintains high similarity with the longer one to retain its desired properties, while also resembling a realistic, relatively shorter sequence to ensure safety and predictability. Algorithm 2 in Appendix E.5 describes using GFNSeqEditor to combine two sequences with the goal of shortening the longer one.

We evaluate GFNSeqEditor's performance in combining pairs of long and short sequences using the AMP dataset as a test case. In this context, a *long sequence* is defined as one with a length exceeding

30, while a *short sequence* has a length shorter than 20. Each initial pair consists of a long AMP sequence and its closest short sequence with an AMP property exceeding 0.7. Table 5 and Figure 5 in Appendix E.5 present the results of sequence combination for sequence length reduction. As indicated in Table 5, GFNSeqEditor not only enhances the properties of the initial long sequences, but also significantly shortens them by more than 63%. Additionally, the sequences generated by GFNSeqEditor resemble both the initial long and short sequences, with an average Levenshtein similarity of approximately 65% to long sequences and 55% to short sequences.

## 6 Conclusions

This paper introduces GFNSeqEditor, a generative model for sequence editing built upon GFlowNet. Given an input seed sequence, GFNSeqEditor identifies and edits positions within the input sequence to enhance its property. This paper also offers a theoretical analysis of the properties of edited sequences and the amount of edits performed by GFNSeqEditor. Experimental evaluations using real-world DNA and protein datasets demonstrate that GFNSeqEditor outperforms state-of-the-art baselines in terms of property enhancement while maintaining a similar amount of edits. Nevertheless, akin to many machine learning algorithms, GFNSeqEditor does have its limitations. It relies on a well-trained GFlowNet model, necessitating the availability of a high-quality trained GFlowNet for optimal performance.

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

# A  Proof of Theorem 4.1

Let $\boldsymbol{z}$ denotes a sequence with length $T$ generated from scratch using the policy $\pi_F(\cdot)$ as

$$\pi_F(a|\boldsymbol{z}_{:t}) = \frac{F_{\boldsymbol{\theta}}(\boldsymbol{z}_{:t} + a)}{\sum_{a' \in \mathbb{A}} F_{\boldsymbol{\theta}}(\boldsymbol{z}_{:t} + a')}. \tag{13}$$

The expected reward of $\boldsymbol{z}$ can be obtained as

$$R_{F,T} = \mathbb{E}[\boldsymbol{z}] = \sum_{\boldsymbol{w} \in \mathbb{T}_T} \Pr[\boldsymbol{z} = \boldsymbol{w}] R(\boldsymbol{w}) = \sum_{\boldsymbol{w} \in \mathbb{T}_T} \prod_{t=1}^{T} \pi_F(w_t|\boldsymbol{w}_{:t-1}) R(\boldsymbol{w}) \tag{14}$$

where $\mathbb{T}_T$ denotes the set of sequences with length $T$ that can be generated by $F_{\boldsymbol{\theta}}(\cdot)$. The probability that the GFNSeqEditor outputs an arbitrary sequence $\boldsymbol{w} \in \mathbb{T}_T$ given $\boldsymbol{x}$ can be expressed as

$$\Pr[\hat{\boldsymbol{x}} = \boldsymbol{w}|\boldsymbol{x}] = \prod_{t=1}^{T} \Pr[\hat{x}_t = w_t|\hat{\boldsymbol{x}}_{:t-1}, \boldsymbol{x}]. \tag{15}$$

The probability $\Pr[\hat{x}_t = w_t|\hat{\boldsymbol{x}}_{:t-1}, \boldsymbol{x}]$ can be obtained as

$$\Pr[\hat{x}_t = w_t|\hat{\boldsymbol{x}}_{:t-1}, \boldsymbol{x}] = \Pr[D(\hat{\boldsymbol{x}}_{:t-1}; \delta, \sigma) = 1]\pi(w_t|\boldsymbol{w}_{:t-1}) + \Pr[D(\hat{\boldsymbol{x}}_{:t-1}; \delta, \sigma) = 0]\mathbf{1}_{w_t = x_t}$$
$$\geq \Pr[D(\hat{\boldsymbol{x}}_{:t-1}; \delta, \sigma) = 1]\pi(w_t|\boldsymbol{w}_{:t-1}) \tag{16}$$

where $\pi(\cdot)$ defined in (6). According to (6), it can be written that

$$\pi(w_t|\boldsymbol{w}_{:t-1}) \geq (1 - \lambda)\frac{F_{\boldsymbol{\theta}}(\boldsymbol{w}_{:t})}{\sum_{a' \in \mathbb{A}} F_{\boldsymbol{\theta}}(\boldsymbol{w}_{:t-1} + a')} = (1 - \lambda)\pi_F(w_t|\boldsymbol{w}_{:t-1}). \tag{17}$$

Furthermore, according to (4) and (5), we have $D(\hat{\boldsymbol{x}}_{:t-1}; \delta, \sigma) = 1$ if

$$\frac{F_{\boldsymbol{\theta}}(\hat{\boldsymbol{x}}_{:t-1} + x_t)}{\sum_{a' \in \mathbb{A}} F_{\boldsymbol{\theta}}(\hat{\boldsymbol{x}}_{:t-1} + a')} - \delta \max_{a \in \mathbb{A}} \frac{F_{\boldsymbol{\theta}}(\hat{\boldsymbol{x}}_{:t-1} + a)}{\sum_{a' \in \mathbb{A}} F_{\boldsymbol{\theta}}(\hat{\boldsymbol{x}}_{:t-1} + a')} < \nu. \tag{18}$$

In addition, it can be inferred that

$$\frac{F_{\boldsymbol{\theta}}(\hat{\boldsymbol{x}}_{:t-1} + x_t)}{\sum_{a' \in \mathbb{A}} F_{\boldsymbol{\theta}}(\hat{\boldsymbol{x}}_{:t-1} + a')} - \delta \max_{a \in \mathbb{A}} \frac{F_{\boldsymbol{\theta}}(\hat{\boldsymbol{x}}_{:t-1} + a)}{\sum_{a' \in \mathbb{A}} F_{\boldsymbol{\theta}}(\hat{\boldsymbol{x}}_{:t-1} + a')} \leq 1 - \delta. \tag{19}$$

Therefore, it can be concluded that if $\nu > 1 - \delta$, it is guaranteed that $D(\hat{\boldsymbol{x}}_{:t-1}; \delta, \sigma) = 1$. Since $\nu$ follows a Gaussian distribution with a variance of $\sigma^2$ we have $\nu > 1 - \delta$ with probability $1 - \Phi(\frac{1-\delta}{\sigma})$. Hence, it can be written that

$$\Pr[D(\hat{\boldsymbol{x}}_{:t-1}; \delta, \sigma) = 1] \geq 1 - \Phi(\frac{1 - \delta}{\sigma}). \tag{20}$$

Combining (20) and (17) with (16), we get

$$\Pr[\hat{x}_t = w_t|\hat{\boldsymbol{x}}_{:t-1}, \boldsymbol{x}] \geq (1 - \lambda)\pi_F(w_t|\boldsymbol{w}_{:t-1})\left(1 - \Phi(\frac{1 - \delta}{\sigma})\right). \tag{21}$$

Moreover, combining (21) with (15), we obtain

$$\Pr[\hat{\boldsymbol{x}} = \boldsymbol{w}|\boldsymbol{x}] \geq \prod_{t=1}^{T} (1 - \lambda)\pi_F(w_t|\boldsymbol{w}_{:t-1})\left(1 - \Phi(\frac{1 - \delta}{\sigma})\right). \tag{22}$$

Using (22) for the expected reward of $\hat{\boldsymbol{x}}$ given $\boldsymbol{x}$ we can write

$$\mathbb{E}[R(\hat{\boldsymbol{x}})|\boldsymbol{x}] = \sum_{\boldsymbol{w} \in \mathbb{T}_T} \Pr[\hat{\boldsymbol{x}} = \boldsymbol{w}|\boldsymbol{x}] R(\boldsymbol{w})$$

$$\geq \sum_{\boldsymbol{w} \in \mathbb{T}_T} \prod_{t=1}^{T} (1 - \lambda)\pi_F(w_t|\boldsymbol{w}_{:t-1})\left(1 - \Phi(\frac{1 - \delta}{\sigma})\right) R(\boldsymbol{w}). \tag{23}$$

Combining (23) with (14), we get

$$\mathbb{E}[R(\hat{\boldsymbol{x}})|\boldsymbol{x}] \geq (1 - \lambda)\left(1 - \Phi(\frac{1 - \delta}{\sigma})\right) R_{F,T} \tag{24}$$

which proves (8). Moreover, the upper bound of property improvement by the proposed GFNSeqEditor is analyzed in Appendix B.

## B   Proof of Theorem 4.2

The expected property improvement of GFNSeqEditor can be obtained as

$$\mathbb{E}[\text{PI}|\boldsymbol{x}] = \sum_{\boldsymbol{w} \in \mathbb{T}_T} \Pr[\hat{\boldsymbol{x}} = \boldsymbol{w}|\boldsymbol{x}](p_{\boldsymbol{w}} - y). \tag{25}$$

Since $\mathbb{T}_T$ can be split into two sets $\mathbb{S}_{\boldsymbol{x}}^*$ and $\mathbb{T}_T \setminus \mathbb{S}_{\boldsymbol{x}}^*$, the expected property improvement of GFNSeqEditor can be obtained as

$$\mathbb{E}[\text{PI}|\boldsymbol{x}] = \sum_{\boldsymbol{w} \in \mathbb{S}_{\boldsymbol{x}}^*} \Pr[\hat{\boldsymbol{x}} = \boldsymbol{w}|\boldsymbol{x}](p_{\boldsymbol{w}} - y) + \sum_{\boldsymbol{w} \in \mathbb{T}_T \setminus \mathbb{S}_{\boldsymbol{x}}^*} \Pr[\hat{\boldsymbol{x}} = \boldsymbol{w}|\boldsymbol{x}](p_{\boldsymbol{w}} - y). \tag{26}$$

If $\boldsymbol{w} \in \mathbb{T}_T \setminus \mathbb{S}_{\boldsymbol{x}}^*$, then $p_{\boldsymbol{w}} \le y$. Therefore, the expected property improvement of GFNSeqEditor can be bounded from above as

$$\mathbb{E}[\text{PI}|\boldsymbol{x}] \le \sum_{\boldsymbol{w} \in \mathbb{S}_{\boldsymbol{x}}^*} \Pr[\hat{\boldsymbol{x}} = \boldsymbol{w}|\boldsymbol{x}](p_{\boldsymbol{w}} - y). \tag{27}$$

The probability that the GFNSeqEditor outputs $\boldsymbol{w} \in \mathbb{S}_{\boldsymbol{x}}^*$ can be expressed as

$$\Pr[\hat{\boldsymbol{x}} = \boldsymbol{w}|\boldsymbol{x}] = \prod_{t=1}^{T} \Pr[\hat{x}_t = w_t|\hat{\boldsymbol{x}}_{:t-1}, \boldsymbol{x}]. \tag{28}$$

The probability $\Pr[\hat{x}_t = w_t|\hat{\boldsymbol{x}}_{:t-1}, \boldsymbol{x}]$ can be obtained as

$$\begin{aligned}\Pr[\hat{x}_t = w_t|\hat{\boldsymbol{x}}_{:t-1}, \boldsymbol{x}] = &\Pr[D(\hat{\boldsymbol{x}}_{:t-1}; \delta, \sigma) = 1]\pi(w_t|\boldsymbol{w}_{:t-1}) \\ &+ \Pr[D(\hat{\boldsymbol{x}}_{:t-1}; \delta, \sigma) = 0]\mathbf{1}_{x_t = w_t}.\end{aligned} \tag{29}$$

If $x_t \ne w_t$, according to (37) and considering the fact that $\pi(w_t|\boldsymbol{w}_{:t-1}) \le 1$, the probability in (29) can be bounded from above as

$$\Pr[\hat{x}_t = w_t|\hat{\boldsymbol{x}}_{:t-1}, \boldsymbol{x}] \le 1 - \Phi(-\frac{\delta}{\sigma}). \tag{30}$$

Otherwise if $x_t = w_t$, it can be written that $\Pr[\hat{x}_t = w_t|\hat{\boldsymbol{x}}_{:t-1}, \boldsymbol{x}] \le 1$. Since any $\boldsymbol{w} \in \mathbb{S}_{\boldsymbol{x}}^*$ should be different from $\boldsymbol{x}$ in at least one position, combining (28) with (30) we can conclude that

$$\Pr[\hat{\boldsymbol{x}} = \boldsymbol{w}|\boldsymbol{x}] \le 1 - \Phi(-\frac{\delta}{\sigma}). \tag{31}$$

Combining (27) with (31) proves the Theorem.

## C   Proof of Theorem 4.3

We obtain the upper bound for the expected distance between edited sequence $\hat{\boldsymbol{x}}$ and the original sequence $\boldsymbol{x}$. Since both $\boldsymbol{x}$ and $\hat{\boldsymbol{x}}$ have the same length $T$, the distance $\text{lev}(\boldsymbol{x}, \hat{\boldsymbol{x}})$ can be interpreted as the number of elements different in these two sequences. Therefore, in order to obtain $\text{lev}(\boldsymbol{x}, \hat{\boldsymbol{x}})$, it is sufficient to find the number of times that $x_t \ne \hat{x}_t, \forall t : 1 \le t \le T$. If $D(\hat{\boldsymbol{x}}_{:t-1}; \delta, \sigma) = 0$, then $\hat{x}_t = x_t$. Furthermore, if $D(\hat{\boldsymbol{x}}_{:t-1}; \delta, \sigma) = 1$, then according to (6), we have $\hat{x}_t = x_t$ with probability

$$(1 - \lambda)\frac{F_{\boldsymbol{\theta}}(\hat{\boldsymbol{x}}_{:t-1} + x_t)}{\sum_{a' \in \mathbb{A}} F_{\boldsymbol{\theta}}(\hat{\boldsymbol{x}}_{:t-1} + a')} + \lambda. \tag{32}$$

Therefore, the probability $\Pr[\hat{x}_t \ne x_t]$ can be obtained as

$$\Pr[\hat{x}_t \ne x_t] = \Pr[D(\hat{\boldsymbol{x}}_{:t-1}; \delta, \sigma) = 1](1 - \lambda)\left(1 - \frac{F_{\boldsymbol{\theta}}(\hat{\boldsymbol{x}}_{:t-1} + x_t)}{\sum_{a' \in \mathbb{A}} F_{\boldsymbol{\theta}}(\hat{\boldsymbol{x}}_{:t-1} + a')}\right). \tag{33}$$

Since $F_{\boldsymbol{\theta}}(\hat{\boldsymbol{x}}_{:t-1} + x_t) \ge 0$, the probability $\Pr[\hat{x}_t \ne x_t]$ can be bounded as

$$\Pr[\hat{x}_t \ne x_t] \le \Pr[D(\hat{\boldsymbol{x}}_{:t-1}; \delta, \sigma) = 1](1 - \lambda). \tag{34}$$

Moreover, if we have

$$\nu \le \frac{F_{\boldsymbol{\theta}}(\hat{\boldsymbol{x}}_{:t-1} + x_t)}{\sum_{a' \in \mathbb{A}} F_{\boldsymbol{\theta}}(\hat{\boldsymbol{x}}_{:t-1} + a')} - \delta \max_{a \in \mathbb{A}} \frac{F_{\boldsymbol{\theta}}(\hat{\boldsymbol{x}}_{:t-1} + a)}{\sum_{a' \in \mathbb{A}} F_{\boldsymbol{\theta}}(\hat{\boldsymbol{x}}_{:t-1} + a')} \tag{35}$$

then $D(\hat{\boldsymbol{x}}_{:t-1}; \delta, \sigma) = 0$. Furthermore, the right hand side of (35) can be bounded from below as

$$\frac{F_{\boldsymbol{\theta}}(\hat{\boldsymbol{x}}_{:t-1} + x_t)}{\sum_{a' \in \mathbb{A}} F_{\boldsymbol{\theta}}(\hat{\boldsymbol{x}}_{:t-1} + a')} - \delta \max_{a \in \mathbb{A}} \frac{F_{\boldsymbol{\theta}}(\hat{\boldsymbol{x}}_{:t-1} + a)}{\sum_{a' \in \mathbb{A}} F_{\boldsymbol{\theta}}(\hat{\boldsymbol{x}}_{:t-1} + a')} \ge -\delta. \tag{36}$$

Therefore, if $\nu \le -\delta$, it is ensured that $D(\hat{\boldsymbol{x}}_{:t-1}; \delta, \sigma) = 0$. The probability that $\nu \le -\delta$ is $\Phi(-\frac{\delta}{\sigma})$. Hence, we can conclude that

$$\Pr[D(\hat{\boldsymbol{x}}_{:t-1}; \delta, \sigma) = 1] = 1 - \Pr[D(\hat{\boldsymbol{x}}_{:t-1}; \delta, \sigma) = 0] \le 1 - \Phi(-\frac{\delta}{\sigma}). \tag{37}$$

Combining (37) with (34), we arrive at

$$\Pr[\hat{x}_t \ne x_t] \le \left(1 - \Phi(-\frac{\delta}{\sigma})\right)(1 - \lambda). \tag{38}$$

Moreover, since both $\boldsymbol{x}$ and $\hat{\boldsymbol{x}}$ have the same length $T$, the expected Levenshtein distance between $\boldsymbol{x}$ and $\hat{\boldsymbol{x}}$ can be obtained as

$$\mathbb{E}[\text{lev}(\boldsymbol{x}, \hat{\boldsymbol{x}})] = \sum_{t=1}^{T} \Pr[\hat{x}_t \ne x_t]. \tag{39}$$

Thus, combining (39) with (38), we can write that

$$\mathbb{E}[\text{lev}(\boldsymbol{x}, \hat{\boldsymbol{x}})] \le \left(1 - \Phi(-\frac{\delta}{\sigma})\right)(1 - \lambda)T \tag{40}$$

which proves the Theorem.

## D    Proof of Theorem 4.4

According to (33) and the assumption in (11), it can be written that

$$\Pr[\hat{x}_t \ne x_t] \ge \Pr[D(\hat{\boldsymbol{x}}_{:t-1}; \delta, \sigma) = 1](1 - \lambda)\epsilon. \tag{41}$$

Combining (41) with (20), we get

$$\Pr[\hat{x}_t \ne x_t] \ge \epsilon(1 - \lambda)\left(1 - \Phi(\frac{1 - \delta}{\sigma})\right). \tag{42}$$

Summing (42) over all elements in the sequence proves the theorem.

## E    Supplementary Experimental Results and Details

This appendix provides a comprehensive overview of the experimental setup in Section 5 and presents additional supplementary experimental results.

### E.1    Implementation Details

All training and inferences, including GFNSeqEditor, have been conducted using a single Nvidia Quadro P6000.

#### E.1.1    Datasets

Detailed information about the datasets can be found below:

- **TFbinding:** The dataset is taken from [4] and contains all possible DNA sequences with length 8. The vocabulary is the four DNA bases, {A, C, G, T}. The goal is to edit a given DNA sequence to increase its binding activity with certain DNA-binding proteins called transcription factors. Higher binding activity is preferable. For train, test and validation purposes 50% of the dataset is set aside. The task entails editing a test dataset consisting of 10% of samples while the remaining data is utilized for training and validation.

- **AMP:** The dataset, acquired from DBAASP [40], is curated following the approach outlined by [19]. Peptides (i.e. short proteins) within a sequence-length range of 12 to 60 amino acids are specifically chosen. The dataset comprises a total of 6,438 positive samples, representing anti-microbial peptides (AMPs), and 9,522 negative samples, which are non-AMPs. The vocabulary consists of 20 amino acids. The primary objective is to edit the non-AMP samples in such a way that the edited versions attain the characteristics exhibited by AMP samples. The task primarily centers on editing a subset comprising 10% of the non-AMP samples, designated for use as test samples, with the remaining samples allocated for training and validation purposes.

- **CRE:** The dataset contains putative human cis-regulatory elements (CRE) which are regulatory DNA sequences modulating gene expression. CREs were profiled via massively parallel reporter assays (MPRAs)[12] where the activity is measured as the expression of the reporter gene. For our analysis, we randomly extract 10,000 DNA sequences, each with a length of 200 base pairs, utilizing a vocabulary of the four bases. The overarching objective is to edit the DNA sequences to increase the reporter gene's expression specifically within the K562 cell line, which represents erythroid precursors in leukemia. The task involves editing a subset of 1,000 test samples, while the rest are allocated for training and validation purposes.

### E.1.2  Oracles

To evaluate the performance of each sequence editing method in terms of property improvement, it is required to obtain the properties of edited sequences. To this end, we employ an oracle for each dataset. The TFbinding dataset contains all possible 65,792 DNA sequences with length of 8. Therefore, by looking into the dataset the true label of each edited sequence can be found. Following [1, 19], the AMP dataset is split into two parts: $D_1$ and $D_2$. The oracle for the AMP dataset is a set of trained models on partition $D_2$ as a simulation of wet-lab experiments. We employed oracles trained by [19] for AMP dataset. It is worth noting that the performance of predictive models on AMP datasets can be influenced by negative sampling in the dataset [44]. Furthermore, for CRE dataset we leverage the Malinois model [12] which is a deep convolutional neural network (CNN) for cell type-informed CRE activity prediction of any arbitrary sequence.

### E.1.3  Baselines Implementation

**DE and Ledidi.** In order to implement DE and Ledidi baselines, there should be a proxy model to enable baselines to evaluate their candidate edits at each iteration of these algorithms. For each dataset, we train a proxy model on the training split of each dataset. For the TFBinding dataset, we configure a three-layer MLP with hidden dimensions of 64. In the case of AMP, we opt for a four-layer MLP, also with hidden dimensions of 64. Finally, for CRE, we utilize a four-layer MLP with hidden dimensions set to 2048. Across all models, the learning rate is consistently set to $10^{-4}$, ReLU serves as the activation function, and we set the number of epochs as 2,000. To implement the DE baseline, we randomly select edit locations based on the desired edit percentages. At each selected location, we apply an edit by choosing the action that maximizes the proxy model's property prediction.

**LaMBO.** We utilize the official implementation of LaMBO[4]. Test sequences[5] serve as the candidate pool, and all candidate samples in the pool are weighted similarly to maintain consistency with other sequence editing baselines. To ensure a fair comparison, we employ the same proxy as GFNSeqEditor to calculate the property scores. Across all three datasets, we use `mlm` as the encoder objective, `ei` as the acquisition function, and DKL SVGP regression as the surrogate. To generate 10 edits per sample, we configure `num-gens=10`, and `window-size` is adjusted for each dataset to ensure the edited

---

[4]https://github.com/samuelstanton/lambo.
[5]For the AMP dataset, we have removed the samples with a length lower than `window-size`.

percentages closely match the desired values. Hence, we set it to 2, 14, and 23 for TFBinding, AMP, and CRE, respectively. Additionally, for all datasets, we set `pref-cond=False`, `pref-alpha=1`, and `beta-sched=1`.

**MOGFN-AL.** We utilize the official implementation of MOGFN-AL[6], where sequence tasks are based on the LaMBO implementation. Similar to LaMBO, test sequences serve as the candidate pool, weighted equally. The same proxy as GFNSeqEditor and LaMBO is employed to calculate property scores. Like LaMBO, we utilize `mlm` as the encoder objective. Training and validation batch sizes are set to 16 and 64, respectively, with hyperparameters for the conditional transformer as follows: `num-hid=128`, `num-layers=3`, and `num-head=8`. For each dataset, we adjust the `max-len` parameter to closely match the desired edited percentage. Thus, for TFBinding, AMP, and CRE, we set them as 6, 30, and 50, respectively. The reward function is modified to match that of GFNSeqEditor. To ensure fair comparisons and adapt LaMBO and MOGFN-AL for sequence editing, we do not use active learning settings for either baseline.

**Seq2Seq.** In order to implement Seq2Seq baseline we use a standard transformer [53] as the translator to map an input sequence to an output sequence with superior property. We paired samples in each dataset such that each pair contains a sequence with lower property and a similar sequence with higher property which is the most similar to the sequence with lower property in the dataset. The transformer is trained to map the low property sequence to the high property sequence in each pair. The transformer is trained using the standard configurations in Pytorch transformer module tutorial. Both the embedding dimension of the transformer and the dimension of the 2 layer feedforward network model in the transformer encoder are set to 200. The number of heads in multihead attention layer is 2 and the drop-out rate is $0.2$. We employ the CrossEntropyLoss function in conjunction with the stochastic gradient descent optimizer. The initial learning rate is set at $5.0$ and follows a StepLR schedule.

### E.1.4   GFlowNet Training

Both the baseline GFlowNet-E and the proposed GFNSeqEditor use the same trained GFlowNet model. We trained an active learning based GFlowNet model following the setting in [19]. In active learning setting, at each round of active learning $t \times K$ candidates generated by GFlowNet are sampled and then top $K$ samples based on scores given by a proxy are chosen to be added to the offline dataset. Here offline dataset refers to an initial labeled dataset. To train the GFlowNet, we employed the same proxy models as those used by other baseline methods. For all datasets, we set the number of active learning rounds to 1, with $t$ equal to 5 and $K$ equal to 100. We parameterize the flow using a MLP comprising two hidden layers, each with a dimension of 2048, and $|\mathbb{A}|$ outputs corresponding to individual actions. Throughout our experiments, we employ the trajectory balance objective for training. Adam optimizer with $(\beta_0, \beta_1) = (0.9, 0.999)$ is utilized during the training process. The learning rate for $logZ$ in trajectory balance loss is set to $10^{-3}$ for all the experiments. The number of training steps for TFbinding, AMP and CRE are 5000, $10^6$ and $10^4$, respectively. The remaining hyperparameters were configured in accordance with the settings established in [19].

### E.2   Diffusion Model Training

We trained our diffusion models on the full sequence datasets of AMP sequences or CRE sequences. The sequences were one-hot encoded, yielding 20-vectors for protein sequences and 4-vectors for DNA sequences.

We employed the "variance-preserving stochastic differential equation" (VP-SDE) [47]. We used a variance schedule of $\beta(t) = 0.9t + 0.1$. We set our time horizon $T = 1$ (i.e. $t \in [0, 1)$). This amounts to adding Gaussian noise over continuous time.

For our discrete-time diffusion model, we defined a discrete-time Gaussian noising process, following [17]. We defined $\beta_t = (1 \times 10^{-4}) + (1 \times 10^{-5})t$. We set our time horizon $T = 1000$ (i.e. $t \in [0, 1000]$).

Our denoising network was based on a transformer architecture. The time embedding was computed as $[\sin(2\pi \frac{t}{T} z), \cos(2\pi \frac{t}{T} z)]$, where $z$ is a 30-vector of Gaussian-distributed parameters that are not trainable. The time embeddings were passed through two dense layers with a sigmoid in between,

---

[6]https://github.com/MJ10/mogfn-al.

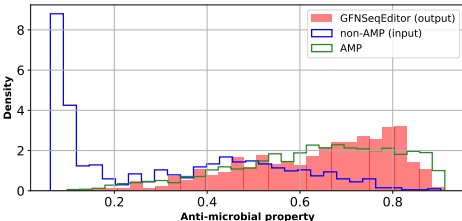

Figure 6: GFNSeqEditor shifts the distribution of non-AMP inputs to the known AMPs.

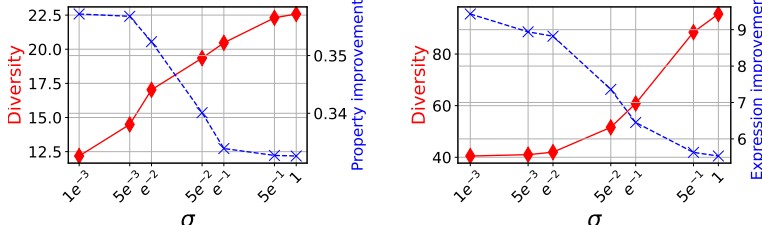

Figure 7: Studying the effect of hyperparameter $\sigma$ on the diversity and performance of GFNSeqEditor over AMP **(left)** and CRE **(right)** datasets.

mapping to a 256-vector of time representations. For any input in a batch, it was concatenated with the time embedding and a sinusoidal positional embedding (defined in [53]) of dimension 64. This concatenation was passed to a linear layer to map it to 128 dimensions. This was then passed to a standard transformer encoder of 3 layers, 8 attention heads, and with a hidden dimension of 128 and an MLP dimension of 64. The result was then passed to a linear layer which projected back to the input dimension.

We trained our diffusion models with a learning rate of $0.001$, for $100$ epochs. We noted that the loss had converged for all models at that point. We also employed empirical loss weighting, where the loss of each input in a batch is divided by the L2 norm of the true Stein score.

We trained our diffusion models on a single Nvidia Quadro P6000.

When generating samples from a continuous-time diffusion model, we used the predictor-corrector algorithm defined in [47], using 1000 time steps from $T$ to 0. We then rounded all outputs to the nearest integer to recover the one-hot encoded sample.

### E.3 Supplementary Results for Sequence Editing

In Figure 6, we illustrate the distribution of input non-AMP sequences, the sequences edited by GFNSeqEditor, and the AMP samples from the AMP dataset. It is evident from Figure 6 that GFNSeqEditor shifts the property distribution of input non-AMP sequences towards that of AMP sequences. Moreover, Figure 7 illustrates the impact of changing $\sigma$ on property improvement and edit diversity for GFNSeqEditor. As can be seen increasing $\sigma$ results in decreased property improvement and enhanced diversity.

It is worth noting that GFNSeqEditor is capable of performing edits even when certain portions of the input sequence are masked and cannot be modified. Table 3 showcases the performance of GFNSeqEditor compared to Ledidi on the CRE dataset, with the first 100 elements of the input sequences masked. As depicted in Table 3, GFNSeqEditor achieves significantly greater property improvement than Ledidi while utilizing a lower edit percentage.

### E.4 Supplementary Results for Sequence Generation

This Subsection compares the performance of GFNSeqEditor in sequence generation task with that of GFlowNet and diffusion model (DM) on CRE dataset. We relax the hyperparameters to allow a higher amount of edits and we set $\delta = 0.4$, $\lambda = 0.1$ and $\sigma = 0.001$ for GFNSeqEditor. The results are presented in Table 4. GFlowNet and DM generate 1000 sequences. GFNSeqEditor also generates 1000 sequences by editing each of 1000 samples in the test dataset. As can be seen, GFNSeqEditor

Table 3: Performance of GFNSeqEditor and Ledidi with 100 elements of each sequence masked for editing for CRE dataset.

| Algorithms | PI | EP(%) | Diversity | PI | EP(%) | Diversity |
|---|---|---|---|---|---|---|
| Ledidi | 0.52 | 18.69 | 38.34 | 0.26 | 14.39 | 37.45 |
| GFNSeqEditor | **4.79** | **17.89** | 32.30 | **4.05** | **14.19** | 25.52 |

Table 4: Performance of GFNSeqEditor, GFlowNet and DM on generating new sequences for CRE dataset.

| Algorithms | Property | Diversity | Distance(%) |
|---|---|---|---|
| DM | 1.75 | 107.38 | 63.59 |
| GFlowNet | 28.20 | 83.88 | 54.41 |
| GFNSeqEditor | **29.25** | 87.32 | **47.34** |

achieves higher property than both GFlowNet and DM. It is useful to note that the experimental study by [19] have shown that GFlowNet outperforms state-of-the-art sequence design methods. For each sequence generated by GFlowNet and DM, the distance to the test set is measured as the distance between the generated sequence and its closest counterpart in the test set. On average, the distance between sequences generated by GFlowNet and the test set is $54.34\%$, while for DM, it is $63.59\%$. GFNSeqEditor achieves superior performance by editing, on average, $47.34\%$ of a sequence in the test dataset. The distance between test set and generated sequences by GFlowNet and DM cannot be controlled. As it is studied in Figures 3 and 7, the amounts of edits performed by GFNSeqEditor can be controlled by hyperparameters $\delta$, $\lambda$ and $\sigma$.

### E.5 Supplementary Discussion and Results for Sequence Combination

Algorithm 2 presents the GFNSeqEditor for combining two sequences in order to obtain a new sequence whose length is the length of shorter sequence. In Figure 5, we depict the distributions of input sequence lengths and properties, alongside the lengths and properties of the outputs generated by GFNSeqEditor. This scenario pertains to the combination of a long AMP sequence with a short AMP sequence, as detailed in Subsection 5.3. As depicted in Figure 5, the edited sequences produced by GFNSeqEditor exhibit property distributions akin to those of the lengthy input AMP sequences. Simultaneously, these edited sequences are considerably shorter than the original long input sequences. This highlights GFNSeqEditor's effectiveness in shortening lengthy AMP sequences while preserving their inherent properties.

Furthermore, Table 6 provides results for combining pairs of AMP sequences as well as pairs consisting of an AMP sequence and a non-AMP sequence. In both cases, GFNSeqEditor generates a sequence with a length matching that of the longer sequence. When combining two AMP sequences, GFNSeqEditor produces new sequences with higher properties than their parent sequences, maintaining an average resemblance of over $60\%$ to each parent. Additionally, GFNSeqEditor can be applied to combine a non-AMP sequence with an AMP sequence, offering the advantage of rendering the edited non-AMP sequence more akin to a genuine AMP sequence. The results in Table 6 demonstrate that GFNSeqEditor substantially enhances the properties of non-AMP sequences, surpassing the properties of their AMP parents. Furthermore, on average, $35\%$ of the edited sequences bear a resemblance to their AMP parents.

## F   Supplementary Related Works

GFlowNets, initially proposed by [5], were introduced as a reinforcement-learning (RL) algorithm designed to expand upon maximum-entropy RL, effectively handling scenarios with multiple paths leading to a common state. However, recent studies have redefined and generalized its scope, describing it as a general framework for amortized inference with neural networks [32, 20, 60, 56].

There has been a recent surge of interest in employing GFlowNets across various domains. Noteworthy examples include its utilization in molecule discovery [5], Bayesian structure learning [11, 36], and

**Algorithm 2** GFNSeqEditor for combining two sequences to shorten the length of longer sequence.

1: **Input:** $\boldsymbol{x}_1$ and $\boldsymbol{x}_2$ with lengths $T_1$ and $T_2$, flow function $F_{\boldsymbol{\theta}}(\cdot)$ and parameters $\delta$, $\lambda$ and $\sigma$.
2: Initialize $\hat{\boldsymbol{x}}_{:0}$ as an empty sequence and $T_{\min} = \min\{T_1, T_2\}$.
3: **for** $t = 1, \ldots, T_{\min}$ **do**
4:     Assign $x_t = \arg\max_{\{x_{1,t}, x_{2,t}\}} \{F_{\boldsymbol{\theta}}(\hat{\boldsymbol{x}}_{:t-1} + x_{1,t}), F_{\boldsymbol{\theta}}(\hat{\boldsymbol{x}}_{:t-1} + x_{2,t})\}$.
5:     Check if $x_t$ is sub-optimal by obtaining $D(x_t, \hat{\boldsymbol{x}}_{:t-1}; \delta, \sigma)$ according to (5).
6:     **if** $D(x_t, \hat{\boldsymbol{x}}_{:t-1}; \delta, \sigma) = 1$ **then**
7:         Sample $\hat{x}_t$ according to policy $\pi(\cdot | \hat{\boldsymbol{x}}_{:t-1})$ as follows:
8:         **if** $T_1 > T_2$ **then**
9:             $\pi(a | \hat{\boldsymbol{x}}_{:t-1}) = (1 - \lambda) \frac{F_{\boldsymbol{\theta}}(\hat{\boldsymbol{x}}_{:t-1} + a)}{\sum_{a' \in \mathbb{A}} F_{\boldsymbol{\theta}}(\hat{\boldsymbol{x}}_{:t-1} + a')} + \lambda \mathbf{1}_{a = x_{1,t}}$.
10:         **else**
11:             $\pi(a | \hat{\boldsymbol{x}}_{:t-1}) = (1 - \lambda) \frac{F_{\boldsymbol{\theta}}(\hat{\boldsymbol{x}}_{:t-1} + a)}{\sum_{a' \in \mathbb{A}} F_{\boldsymbol{\theta}}(\hat{\boldsymbol{x}}_{:t-1} + a')} + \lambda \mathbf{1}_{a = x_{2,t}}$.
12:         **end if**
13:     **else**
14:         Assign $\hat{x}_t = x_t$.
15:     **end if**
16: **end for**
17: **Output:** Edited sequence $\hat{\boldsymbol{x}}$.

Table 5: Performance of GFNSeqEditor for sequence reduction on AMP dataset in terms of variation in property, edit percentage of long sequences (EPLS), edit percentage of short sequences (EPSS), and percentage of length reduction in the long sequences.

| Input Property | Output Property | EPLS(%) | EPSS(%) | Sequence Reduction(%) |
|---|---|---|---|---|
| 0.65 | 0.67 | 35.96 | 44.65 | 63.23 |

graph explainability [29]. Recognizing its significance, several studies have emerged to enhance the learning efficiency of GFlowNets [6, 32, 30, 43] since the introduction of the flow matching learning objective by [5]. Moreover, GFlowNets have demonstrated adaptability in being jointly trained with energy and reward functions [57]. [38] introduce intrinsic exploration rewards into GFlowNets, addressing exploration challenges within sparse reward tasks. A couple of recent studies try to extend GFlowNets to stochastic environments, accommodating stochasticity in transition dynamics [39] and rewards [58]. Several novel GFlowNet training methodologies have been recently proposed in [22, 26, 37, 14]. The application of GFlowNets when a predefined reward function is not accessible is explored in [8]. Distributed training of GFlowNets is discussed in [45]. Accelerating GFlowNet training is investigated in [25]. Moreover, [27] employs GFlowNets for designing DNA-encoded libraries. To reduce the need for expensive reward evaluations, [24] proposes a new GFlowNet-based method for molecular optimization.

The aforementioned works have primarily focused on theoretical developments of GFlowNet and its application in molecular generation, without directly addressing the challenges associated with sequence design or editing. In a departure from this trend, and inspired by Bayesian Optimization, [19] proposed a new active learning algorithm based on GFlowNets, i.e. GFlowNet-AL to design novel biological sequences. GFlowNet-AL [19] utilizes the epistemic uncertainty of the surrogate model within its reward function, guiding the GFlowNet towards the optimization of promising yet less-explored regions within the state space. This approach fosters the generation of a diverse set of *de novo* sequences from scratch and token-by-token. Unlike GFNSeqEditor, it lacks the capability to edit input seed sequences and combine multiple sequences. This distinction underscores the unique contribution of GFNSeqEditor in addressing the sequence editing problem, positioning it as a valuable addition to the existing literature on GFLowNet.

# G   Societal Impact

Biological sequence optimization and design hold transformative potential for biotechnology and health, offering enhanced therapeutic solutions and a vast range of applications. Techniques that

Table 6: Performance of GFNSeqEditor for sequence combination over AMP dataset in terms of property improvements of first (PI-S$_1$) and second (PI-S$_2$) sequences, edit percentages of first (EP-S$_1$) and second (EP-S$_2$) sequences, and diversity.

| Seq$_1$ | Seq$_2$ | PI-S$_1$ | PI-S$_2$ | EP-S$_1$(%) | EP-S$_2$(%) | Diversity |
|---------|---------|----------|----------|-------------|-------------|-----------|
| AMP | AMP | 0.05 | 0.06 | 36.10 | 39.47 | 7.29 |
| non-AMP | AMP | 0.41 | 0.04 | 41.41 | 65.39 | 12.77 |

enable refining sequences can lead to advancements like elucidating the role of individual gene products, rectifying genetic mutations in afflicted tissues, and optimizing properties of peptides, antibodies, and nucleic-acid therapeutics. However, the dual-edged nature of such breakthroughs must be acknowledged, as the same research might be misappropriated for unintended purposes. Our method can be instrumental in refining diagnostic procedures and uncovering the genetic basis of diseases, which promises a deeper grasp of genetic factors in diseases. Yet, we must approach with caution, as these advancements may unintentionally amplify health disparities for marginalized communities. As researchers, we emphasize the significance of weighing the potential societal benefits against unintended consequences while remaining optimistic about our work's predominant inclination towards beneficial outcomes.

