# OpenReview forum: "GFlowNet Assisted Biological Sequence Editing"
_NeurIPS.cc/2024/Conference — NeurIPS 2024 poster_

### Official Review · Reviewer_Ns1N · 2024-06-22

**Soundness:** 3
**Presentation:** 3
**Contribution:** 3
**Rating:** 6
**Confidence:** 4

**Summary:**

The author proposed GFNSeqEditor, a novel sequence editing and generation model built on GFlowNet, which provides different modifications for each sequence to enhance desired features. Several experiments have demonstrated the performance of the proposed algorithm.

**Strengths:**

A new biological sequence editing method based on GFlowNet has been proposed, capable of identifying and editing positions in a given sequence. It has been demonstrated that the lower and upper bounds on the number of edits performed by GFNSeqEditor can be controlled by adjusting hyperparameters.

**Weaknesses:**

There are shortcomings in the model comparison, as it does not compare with existing biological sequence design methods. In terms of sequence evaluation, mainstream evaluation experiments were not used, leading to doubts about the model's effectiveness.

**Questions:**

1. Existing biological sequence design methods, such as evolutionary approaches, can still perform sequence editing in specified regions. Examples include AdaLead[1], PEX[2], Coms[3], and BiB[4]. The author must compare their method with these state-of-the-art algorithms to demonstrate its advantages.


2. The author used AMP and CRE datasets and trained Oracles independently. Evaluating with self-trained models can lead to inconsistent standards, as the author can modify their trained Oracles at any time to demonstrate the algorithm's superiority. Unlike Coms[3] or BiB[4], which use numerous sequence evaluation problems, this leads to unfair experimental evaluations. It is recommended that the author evaluate under existing various evaluation standards to demonstrate the model's advantages better.


[1] Sinai, Sam, et al. "AdaLead: A simple and robust adaptive greedy search algorithm for sequence design." arXiv preprint arXiv:2010.02141 (2020).

[2] Ren, Zhizhou, et al. "Proximal exploration for model-guided protein sequence design." International Conference on Machine Learning. PMLR, 2022.

[3]  Trabucco, Brandon, et al. "Conservative objective models for effective offline model-based optimization." International Conference on Machine Learning. PMLR, 2021.

[4] Chen, Can, et al. "Bidirectional learning for offline model-based biological sequence design." International Conference on Machine Learning. PMLR, 2023.

**Limitations:**

yes

---

> ### Author Rebuttal · Authors · 2024-08-07
>
> Thank you so much for your review and letting us know your valuable comments. Please find below our responses to your comments.
>
> ## Evolutionary-Based Methods
> We would like to clarify that we have already included the evolutionary method in AdaLead (reference [30] in the paper) among our baselines. As indicated on line 265 of page 7, the DE baseline refers to the evolutionary method presented in [30]. However, instead of naming this baseline as AdaLead, we referred to it as DE, which stands for Directed Evolution. To avoid any confusion, we can revise the name of the DE baseline to AdaLead.
>
> PEX is an evolutionary method that operates through multiple rounds of interactions with the lab. However, these wet lab evaluations can be costly and time-consuming. The focus of this paper is to propose edits without the need for any wet lab interactions. Therefore, we compared the performance of GFNSeqEditor with other baselines that do not have such need. Indeed, if we run PEX only in one iteration without any additional wet lab experiments, the PEX result would be similar to the DE reported in the paper. To address the reviewer's concern, we will add some notes about PEX in our literature review.
>
> ## Model-Based Optimization Methods
> Both Coms and BiB are model-based optimization (MBO) methods. To generate sequences, they perform several rounds of optimization on a set of sequences. However, in biological sequence editing, we often aim to generate sequences similar to a pre-specified seed sequence. Using MBO-based methods for this purpose requires performing optimization on each seed sequence, which makes computations infeasible, especially when it comes to editing thousands of sequences. Moreover, adapting Coms and BiB for sequence editing purposes may require some changes in these algorithms. The advantages of GFNSeqEditor over MBO-based methods are summarized as follows:
> 1. To edit a sequence, GFNSeqEditor employs a pre-trained flow function. GFNSeqEditor only performs inference using the flow function and employing GFNSeqEditor does not involve any model training. This makes the GFNSeqEditor computationally efficient.
> 2. Our paper theoretically proves that using GFNSeqEditor the amount of edits can be controlled while MBO based methods cannot provide such guarantee.
> 3. MBO-based approaches require evaluating the properties of unseen sequences using a proxy model, whereas GFlowNet-based approaches can operate without this. Proxy models may provide misleading predictions for out-of-distribution sequences.
>
> Furthermore, we have already compared GFNSeqEditor with Ledidi, an optimization-based baseline specifically designed for biological sequence editing. To address your concern, we can include background information on BiB in our literature review.
>
> ## Evaluations
> **We did not train oracles for AMP and CRE datasets and the oracles are not self-trained**. For AMP, we employed the oracles used by [14] which can be downloaded from Github repo https://github.com/MJ10/BioSeq-GFN-AL. For CRE, we used the Malinois model [10] which can be obtained from GitHub repo https://github.com/sjgosai/boda2. Therefore, we believe that our comparisons are fair. We clarify this in Appendix E.1.2 on page 16, where we provide detailed information about the oracles used in this paper. Furthermore, we performed experiments on both DNA and protein sequence datasets. Finally, following previous works (see e.g., [10], [14], [23]) and addressing the specific needs of this study, we evaluated the performance of the algorithms based on several important metrics in the biological sequence domain, including property improvement, diversity, and edit percentage.

---

> ### Comment · Reviewer_Ns1N · 2024-08-09
> **Edit Official Comment by Reviewer Ns1N**
>
> Thank you for addressing my concerns and providing detailed explanations in your rebuttal. As a result, I have raised my score.

---

### Official Review · Reviewer_a1PN · 2024-07-08

**Soundness:** 4
**Presentation:** 3
**Contribution:** 3
**Rating:** 7
**Confidence:** 3

**Summary:**

This paper introduced a new algorithm for biological sequence editing, GFNSeqEditor. This algorithm is designed based on pre-trained Generative Flow Networks (GFNs), and improves target properties by identifying and editing sub-optimal sites of input sequences. Through theoretical analysis and experiments on three datasets, this new algorithm shows that it can improve biological properties with diversified edit while minimize the number of edits to ensure safety and predictability. By comparing with a few baseline models, the GFNSeqEditor outperforms the state-of-the-art methods on property improvements, diversity, and edit percentage.

**Strengths:**

The paper includes comprehensive theoretical analysis of the algorithm, such as bounds on expected reward, property improvements, and the number of edits. Together with extensive experiments, this paper provides sufficient information on the influence of hyperparameter choices on algorithm performance, which offers valuable insights for downstream studies.
This paper shows the versatility of GFNSeqEditor through examples of sequence generation and sequence length reduction, indicating that this algorithm can not only be used on sequence editing but also in a broader potential applications in synthetic biology.

**Weaknesses:**

1. the algorithm was built on the assumption that fewer edits lead to safer and more predictable modification. I would argue this claim is valid under the condition that, compared with generating a new sequence, limited editing is possibly a safer choice. Many studies on single nucleotide variance (SNV) show that a single nucleotide change can lead to functional change or property loss. To avoid unexpected confusion, it would be better to either include followup analysis on structural changes caused by sequence editing, which I believe will be too much work to do, or improve the statement on the safety claim.
2. Lack of details about the selection of non-AMP samples. Previous study [1] have shown that inappropriate selection of negative samples can introduce bias. Clarification on the criteria of selecting negative samples would help the audience better evaluate the validity of the results.
3. While GFNSeqEditor shows improvements compared with the baselines, the performance improvements are not substantial, especially on the aspect of editing percentage. Any possible reasons or statistical analysis on the significance?

[1]. Sidorczuk K, Gagat P, Pietluch F, et al. Benchmarks in antimicrobial peptide prediction are biased due to the selection of negative data[J]. Briefings in Bioinformatics, 2022, 23(5): bbac343.

**Questions:**

1. I would suggest refining the statement on fewer edits leads to safe and predictable sequence editing. Follow up analysis on how sequence editing influence structure and function change would be great to have, but not necessary.
2. Please consider add more details on how the dataset select non-AMP samples, and briefly mention about the negative sample problem for clarify.
3. It would be more clear to include statistical analysis or dig into possible reasons and future improvements would help the audience to understand the significance of this work and looking for future improvements.

**Limitations:**

Yes.

---

> ### Author Rebuttal · Authors · 2024-08-07
>
> Thank you very much for taking time to review our paper and let us know your valuable comments. Please find below our responses to your comments and questions.
>
> ## Safety Assumption
> It is generally expected that fewer modifications in biological sequences are less likely to result in significant functional changes or property loss. However, we acknowledge that even limited modifications can lead to functional changes in the edited biological sequence. We agree that the safety assumption has its uncertainties and that investigating structural changes due to sequence editing is beyond the scope of this paper. Therefore, we will revise the safety statement in the paper to reflect this uncertainty.
>
> ## Selection of non-AMP Samples
> For the AMP dataset, we utilized predictive models and data splits from published works [1,14, 27]. To address the reviewer's concern, we will provide additional details about the data sampling process for the AMP dataset in Appendices E.1.1 and E.1.2, using the explanations provided in [1,14, 27]. Furthermore, we will highlight the importance of negative data selection in the performance of predictive models, as discussed in the study by Sidorczuk et al. [R1], suggested by the respected reviewer.
>
> ## Statistical Analysis of Results
> In our experiments, we observed that the property improvements provided by GFNSeqEditor and other baselines depend on the edit percentage. To ensure fair comparisons, we fixed the edit percentage across all algorithms to a similar level whenever possible. This is reflected in Table 1, where the edit percentages for virtually all algorithms are quite similar. According to Table 1, the property improvements provided by GFNSeqEditor are significant for the AMP and CRE datasets compared to other baselines. Furthermore, Figure 2 demonstrates the property improvements of GFNSeqEditor and other baselines as the edit percentage changes. To provide a statistical analysis of the results, Figure 6 in Appendix E.3 illustrates the distribution of properties of edited sequences. We are open to conducting additional statistical analyses based on your suggestions.
>
> ## Reference
> [R1] Sidorczuk K, Gagat P, Pietluch F, et al. Benchmarks in antimicrobial peptide prediction are biased due to the selection of negative data[J]. Briefings in Bioinformatics, 2022, 23(5).

---

### Official Review · Reviewer_6Uc5 · 2024-07-11

**Soundness:** 3
**Presentation:** 2
**Contribution:** 2
**Rating:** 6
**Confidence:** 4

**Summary:**

They propose a new sequence editing method using GFlowNets as priors and suggest additional hyperparameters to tune suboptimal gaps, randomness, and penalization. They theoretically analyze how these new hyperparameters can effectively control the lower and upper bounds of the number of edits. The performance results demonstrate some effectiveness of the proposed method.

**Strengths:**

This approach introduces a new sequence editing method by leveraging the pretrained generative model, GFlowNets, as a prior. I appreciate their motivation for DNA editing and the clear narrative they present. Their theoretical analysis is robust and effectively explains the new hyperparameters.

**Weaknesses:**

**The baselines used in the study are too weak:** I don't think this method offers significant advantages over conditional GFlowNets, which can translate input sequences to output sequences while maintaining constraints on sequence distance. There are many other methods capable of achieving this, including Seq2Seq, which the authors used as a baseline. The Seq2Seq implementation seems overly simplistic; there are potentially better techniques for DNA editing. For example, hierarchical variational inference in latent space, combined with a well-designed reward model, could optimize the process more effectively. Any thoughts on this?

**This is not a GFlowNet with multiple backward paths:** It appears that GFlowNets are used for sequence generation in an autoregressive manner (one-way generation) where \( P_B = 1 \). This approach is equivalent to Soft-Q-learning and path consistency learning (PCL) [1], so relevant soft RL literature should be clearly explained. The paper's title is somewhat misleading (though it need not be changed) as it essentially describes using a PCL-trained agent to assist in sequence editing. Reviewers suggest that the authors acknowledge this literature.

**The literature review for GFlowNets should include works published in 2024:** Reviewers also expect the authors to include recent GFlowNets literature targeting biological or chemical applications, such as those published in ICLR and ICML 2024. Additionally, there are preprints discussing improvements and evolutionary methods (e.g., genetic algorithms) using GFlowNets; engaging with this existing literature would be beneficial. The current literature review seems not very up-to-date in 2024.

[1] Nachum, Ofir, et al. "Bridging the gap between value and policy based reinforcement learning." Advances in neural information processing systems 30 (2017).

**Questions:**

1. There are many hyperparameters, and biological tasks often involve expensive oracle functions. How can we tune these hyperparameters in real-world applications?

2. Is there any tendency for the proposed method to work better on large sequences? Please provide insights on this.

**Limitations:**

They address their limitations in the conclusion.

---

> ### Author Rebuttal · Authors · 2024-08-07
>
> We would like to express our gratitude for taking the time to review our paper and letting us know your thoughtful comments. Please find below our responses to your comments.
>
> ## Conditional GFlowNets
> Conditional GFlowNets can be used for sequence editing by training the flow function with a sequence distance constraint. However, implementing this approach for sequence editing is not scalable and can be computationally intractable. The distance constraint depends on the seed sequence, requiring a separate flow function for each seed sequence to obtain diverse edits. This approach encounters several challenges:
> - Training GFlowNets for each seed sequence is infeasible for large datasets.
> - This approach requires evaluating the properties of unseen sequences using a proxy model, which can lead to inaccurate predictions for out-of-distribution sequences, resulting in poorly trained flow functions.
> - It may reduce the generalizability of the flow function, with the trained flow functions potentially containing only local information.
>
> Moreover, it is uncertain whether such an approach can provide theoretical guarantees on the amount of edits, similar to those provided by the proposed GFNSeqEditor.
>
> In contrast, using GFNSeqEditor, one can train one flow function on offline data. Then for sequence editing, GFNSeqEditor only makes inferences with the flow function. This makes GFNSeqEditor computationally efficient. Moreover, the paper theoretically proves that employing GFNSeqEditor, the amount of edits can be controlled using three hyperparameters.
>
> ## Variational Inference
> In order to compare GFNSeqEditor with variational inference based methods, we include LaMBO [32] as a baseline. LaMBO maps sequences into a latent space and uses Bayesian optimization in this space to generate sequences with relatively higher properties. We find controlling the amount of edits challenging using LaMBO. To sum up, we believe that performing sequence editing with variational inference based methods can be an interesting future research direction as it requires careful and intricate algorithm design.
>
> ## Soft-Q-Learning and PCL Literature Review
> We will acknowledge the relevant literature on Soft-Q-Learning and path consistency learning (PCL). We will add that “Generating sequences in an autoregressive fashion using GFlowNet involves only one path to generate a particular sequence. In such cases, generating biological sequences with GFlowNet can be viewed as a Soft-Q-Learning [R1, R2, R3] and path consistency learning (PCL) [R4] problem.”
>
> ## GFlowNets Literature Review
> We will expand our literature review to include more recent studies on GFlowNets. We will add notes on references [R5]--[R13]. Several novel GFlowNet training methodologies are proposed in [R5, R7, R10, R12]. The application of GFlowNets when a predefined reward function is not accessible is explored in [R6]. Distributed training of GFlowNets is discussed in [R8]. Accelerating GFlowNet training is investigated in [R9]. Moreover, [R11] employs GFlowNets for designing DNA-encoded libraries. To reduce the need for expensive reward evaluations, [R13] proposes a new GFlowNet-based method for molecular optimization.
>
> ## Responses to Questions
> **Response to Question 1**: The theoretical bounds obtained in Section 4.3 can be used to determine a range for hyperparameters. Subsequently, a few experiments can be conducted within this range to identify the optimal hyperparameter values.
>
> **Response to Question 2**: The intuition behind GFNSeqEditor’s superior performance with larger sequences is its utilization of a flow function which is trained to capture global information about the sequence space. In contrast, the performance of local search-based baselines decrease as sequence length increases. This is because, as the search space expands, it becomes more challenging for local search methods to find sequences with optimal performance.
>
> ## References
> [R1] T. Haarnoja, H. Tang, P. Abbeel and S. Levine, “Reinforcement Learning with Deep Energy-Based Policies,” ICML 2017.
>
> [R2] J. Grau-Moya, F. Leibfried and P. Vrancx, “Soft Q-Learning with Mutual-Information Regularization,” ICLR 2019.
>
> [R3] S. Mohammadpour, E. Bengio, E. Frejinger and P. Bacon “Maximum entropy GFlowNets with soft Q-learning,” AISTATS 2024.
>
> [R4] O. Nachum, M. Norouzi, K. Xu and D. Schuurmans “Bridging the Gap Between Value and Policy Based Reinforcement Learning,” NeurIPS 2017.
>
> [R5] H. Jang, M. Kim, S. Ahn, “Learning Energy Decompositions for Partial Inference in GFlowNets,” ICLR 2024.
>
> [R6] Y. Chen and L. Mauch, “Order-Preserving GFlowNets,” ICLR, 2024.
>
> [R7] M. Kim, T. Yun, E. Bengio, D. Zhang, Y. Bengio, S. Ahn and J. Park, “Local Search GFlowNets,” ICLR, 2024.
>
> [R8] T. Silva, L. M. Carvalho, A. H. Souza, S. Kaski and D. Mesquita, “Embarrassingly Parallel GFlowNets,” ICML, 2024.
>
> [R9] M. Kim, J. Ko, T. Yun, D.i Zhang, L. Pan, W. C. Kim, J. Park, E. Bengio and Y. Bengio “Learning to Scale Logits for Temperature-Conditional GFlowNets,” ICML, 2024.
>
> [R10] P. Niu, S. Wu, M. Fan and X. Qian, “GFlowNet Training by Policy Gradients,” ICML, 2024.
>
> [R11] M. Koziarski, M. Abukalam, V. Shah, L. Vaillancourt, D. A. Schuetz, M. Jain, A. van der Sloot, M. Bourgey, A. Marinier and Y. Bengio, “Towards DNA-Encoded Library Generation with GFlowNets,” ICLR 2024 Workshop on Generative and Experimental Perspectives for Biomolecular Design, 2024.
>
> [R12] S. Guo, J. Chu, L. Zhu, Z. Li and T. Li, “Dynamic Backtracking in GFlowNets: Enhancing Decision Steps with Reward-Dependent Adjustment Mechanisms,” Arxiv, 2024.
>
> [R13] H. Kim, M. Kim, S. Choi and J. Park, “Genetic-guided GFlowNets for Sample Efficient Molecular Optimization,” Arxiv, 2024.

---

> > ### Comment · Reviewer_6Uc5 · 2024-08-08
> >
> > Most of my concerns were addressed, so I increased the score accordingly.

---

### Decision · Program_Chairs · 2024-09-25

**Decision:**

Accept (poster)

**Comment:**

This paper presents a novel biological sequence editing algorithm, GFNSeqEditor, based on pretrained Generative Flow Networks (GFNs) with theoretical analysis on expected reward, property improvements, and the number of edits. It identifies and edits sub-optimal sites to enhance the desired property.

Overall, the reviews highlight its strengths of GFNSeqEditor along with theoretical anlyses to understand how it works as well as solid benchmarking evaluations. While the reviewers also raised several potential concerns, most were effectively addressed in the detailed author feedback.

Based on these evaluation, I would like to recommend accepting this submission. We believe this interesting work will attract the interest of the NeurIPS community.